# Trigeminal Sensory Supply Is Essential for Motor Recovery after Facial Nerve Injury

**DOI:** 10.3390/ijms232315101

**Published:** 2022-12-01

**Authors:** Svenja Rink-Notzon, Jannika Reuscher, Klaus Nohroudi, Marilena Manthou, Tessa Gordon, Doychin N. Angelov

**Affiliations:** 1Department of Prosthetic Dentistry, School of Dental and Oral Medicine, University of Cologne, 50931 Cologne, Germany; 2Anatomical Institute II, University of Cologne, 50931 Cologne, Germany; 3Anatomical Institute I, University of Cologne, 50931 Cologne, Germany; 4Department of Histology and Embryology, Aristotle University Thessaloniki, GR-541 24 Thessaloniki, Greece; 5Department of Surgery, The Hospital for Sick Children, Toronto, ON M5G 1X8, Canada

**Keywords:** rat, facial nerve, motoneuron, axotomy, trigeminal nerve, stimulation, vibrissal whisking, motion analysis, recovery of function, morphological correlation

## Abstract

Recovery of mimic function after facial nerve transection is poor. The successful regrowth of regenerating motor nerve fibers to reinnervate their targets is compromised by (i) poor axonal navigation and excessive collateral branching, (ii) abnormal exchange of nerve impulses between adjacent regrowing axons, namely axonal crosstalk, and (iii) insufficient synaptic input to the axotomized facial motoneurons. As a result, axotomized motoneurons become hyperexcitable but unable to discharge. We review our findings, which have addressed the poor return of mimic function after facial nerve injuries, by testing the hypothesized detrimental component, and we propose that intensifying the trigeminal sensory input to axotomized and electrophysiologically silent facial motoneurons improves the specificity of the reinnervation of appropriate targets. We compared behavioral, functional, and morphological parameters after single reconstructive surgery of the facial nerve (or its buccal branch) with those obtained after identical facial nerve surgery, but combined with direct or indirect stimulation of the ipsilateral infraorbital nerve. We found that both methods of trigeminal sensory stimulation, i.e., stimulation of the vibrissal hairs and manual stimulation of the whisker pad, were beneficial for the outcome through improvement of the quality of target reinnervation and recovery of vibrissal motor performance.

## 1. Clinical Importance of the Afferent Fibers, including Trigeminal Nerve Fibers, for Appropriate Motor Nerve Regeneration and Recovery of Motor Performance

The trigeminal and facial nerves are the most frequently affected cranial nerves in maxillofacial trauma [1]. Apart from the rather common idiopathic Bell’s palsy (incidence of 11.5–53.3 per 100,000 individuals a year) [2,3] and traffic accident injuries (brainstem hemorrhage, temporal bone fractures, or lacerations of the face), most facial nerve lesions are postoperative (removal of cerebellopontine angle tumors, acoustic neuroma surgery, or parotid resections because of malignancy). Despite the use of all available microsurgical techniques for repair of transected nerves, the recovery of facial tone, voluntary movement and emotional expression of the face remains poor [4,5,6,7,8]. 

The inevitable occurrence of a “post-surgery paralytic syndrome” including abnormally associated movements and altered blink reflexes [9,10], has been attributed to (1) “misdirected” reinnervation of the facial muscles [11,12], (2) possible ephaptic transmission between axons from adjacent fascicles [13], and (3) putative alterations in synaptic input to facial motoneurons [14,15,16]. 

A key issue in these explanations, which are not mutually exclusive, is the abnormal activity pattern of the axotomized facial motoneurons, which have been isolated from their target by injury. On one hand, they become hyperexcitable [17,18] because of their increased resting membrane potential and the remaining function of their axo-dendritic synapses [19,20]. On the other hand, their reduced synthesis of transmitter-related enzymes [21] and axo-somatic synaptic input [22,23,24,25,26,27] render them less able to respond to afferent input and unable to discharge, resulting in ‘electrical silence’ [28]. Yet, both clinical and experimental data show that functional recovery is generally better after damage to a purely motor nerve than after damage to a mixed (sensory and motor) nerve [29,30,31,32,33]. One possible explanation for the poor recovery is that a lack of sensory input deprives motoneurons of their electrotonic support. Furthermore, motoneuron deafferentation and atrophy occur [27,34,35,36]. Given the extreme disturbances in sensory and motoneuronal excitability following damage to sensory afferents [37], as well as excessive sprouting [38,39] and loss of GABAergic inhibitory control [40], it is not surprising that functional recovery in animals is poor when deafferentation is permanent [25,41]. Interestingly, in the study by Pavlov et al. [41], manual stimulation of the denervated muscles worsened the functional outcome. The finding raises the possibility that the stimulation “overloads” a system that, due to deafferentation, is already hyperexcitable, and possibly leads to irretrievable damage [41]. 

When examining the functional recovery after peripheral nerve injury and surgical repair, we can observe that the nerve supply to facial muscles has the distinct advantage of the motor and sensory nerves being separate [42,43]. Recent work indicates that most cranial nerve motor neurons are typically connected by second order sensory neurons mostly located in the reticular formation [44]. Nevertheless, recalling the rapidness of the trigemino-facial corneal reflex, we cannot exclude that the sensory feedback on the facial motoneurons in the brainstem nucleus occurs (at least partially) via ipsilateral monosynaptic connections on the motoneurons by the trigeminal afferent nerves [45,46,47,48,49,50,51]. The facial nerve injury and surgical repair model that we have used extensively in our studies of nerve injury and recovery has the advantage that the experimenter can observe postoperative vibrissae paralysis, and any recovery of their rhythmical vibrissae whisking, in addition to making a detailed immunohistochemical analysis of the reinnervation of the vibrissal muscles. Moreover, the role of the sensory afferents has been studied by selective extirpation of the infraorbital sensory nerve to the vibrissae. 

In this review, we describe the functional and anatomical outcomes after self-reinnervation of the muscles that mediate whisker movements and the effects of (A) eliminating sensory input from the vibrissal muscle pads to the facial motoneurons by extirpating the ipsilateral infraorbital nerve (IOn, an end branch of the maxillary nerve, which in turn is the second branch of the trigeminal nerve, V) that provides sensory supply to the whisker pad (vibrissal hairs, muscles and fur); and (B) stimulating the IOn directly or indirectly. The research goal(s), the rat surgeries, the outcome measures and data are briefly described and summarized in Table 1, and their significance discussed in each case. We demonstrate that as described by others (see above), electrical silence of axotomized facial motoneurons may be reduced by increasing sensory input from the trigeminal nerve via IOn, the nerve that provides exclusive sensory innervation to the vibrissal muscles [52,53,54]. Our evidence supports the anatomical, electrophysiological, and clinical data that has demonstrated that the trigeminal system is involved in generating facial muscle responses and blink reflexes [42,43].

## 2. Ablation of Trigeminal Sensory Input by Excising the Ipsilateral Infraorbital Nerve (IOn-ipsi-ex) Impedes Recovery of Vibrissae Motor Performance after Transection and Suture of the Facial Nerve (Fn-n). Manual Stimulation (Mstim) of the Paralyzed and Deafferented Whisker Pad Worsens Recovery of Whisking. See Flow Chart Table 2 for a Synopsis

### 2.1. Functional Recovery

The mystacial vibrissae of the rat are normally erect with an anterior orientation. The vibrissae normally move with rostral protraction and caudal retraction by the piloerector muscles, in a highly coordinated rhythmic “whisking” manner with the normal amplitude of ~50° [55,56,57,58] (Figure 1; [59,60,61]). The whisking frequency (6–9 Hz) remained normal two months after facial nerve transection and suture (Fn-n; Figure 2), with and without innervation by the sensory afferents of the IOn, but functional recovery was poor, with a significant fall in the whisking amplitude from mean ± standard deviation (SD) values of 62 ± 13° to 19 ± 6° (intact sensory input) to 22 ± 3° (abolished sensory input by IOn-ex), respectively (Table 2).
ijms-23-15101-t002_Table 2Table 2Experimental design flow chart depicting animal grouping and sequence of the main procedures in the section, describing the effect of the ablation of trigeminal sensory input on the recovery of vibrissae motor performance after transection and suture of the facial nerve. Fn-n, transection and suture of the facial nerve; Ion-ipsi-ex, excision of the ipsilateral infraorbital nerve; Mstim, manual stimulation.Research GoalsDay 0(Beginning)Day 1: Surgeries Day 2–63 (Two Months)Day 64 (8 Weeks Later)Day 65 (1 Day Later) Day 75(10 Days Later)Results**Section 2**: To determine recovery of vibrissae motor performance after transection and suture of the facial nerve (Fn-n), and after Ablation of trigeminal sensory input, by excising the ipsilateral infraorbital nerve (IOn-ipsi-ex).Clipping of all vibrissal hairs except 2 on each side of the face in preparation for the videotaping.Pre-operative videotaping of 24 intact rats for video-based motion analysis of the vibrissae movements.Right (Fn-n) in all 24 rats. 6 rats (group 1) received no other treatment. 6 rats received daily Mstim of the whisker pad (group 2). Another 6 rats received an excision of the ipsilateral infraorbital nerve (IOn-ipsi-ex) (group 3). The last 6 rats received Mstim in addition to Fn-n + IOn-ipsi-ex (group 4). Daily manual stimulation of the whisker pad in groups 2 and 4. Clipping of all vibrissal hairs except 2 on each side of the face in preparation for the videotaping. Postoperative videotaping of all 24 rats for video-based motion analysis of the vibrissae movements.Perfusion fixation with 4% paraformaldehyde in phosphate-buffered saline, pH 7.4 Cutting of the brainstems on a vibratome and determining the intensity of fluorescence after immunostaining for synaptophysin. Cutting of the ipsilateral to Fn-n levator labii superioris muscle (LLS) on a cryostat and determining the degree of polyinnervation of the motor endplates. Amplitude: Intact: 62 ± 6°Fn-n: 19 ± 6°Fn-n + Mstim: 51 ± 19°Fn-n + IOn-ipsi-ex: 22 ± 3° Fn-n + Ion-ipsi-ex + Mstim: 14 ± 6°Polyinnervation %: Intact: 0% Fn-n: 53 ± 10%; Fn-n + Mstim: 22 ± 3%; Fn-n + Ion-ipsi-ex: 43 ± 9%; Fn-n + Ion-ipsi-ex + Mstim: 51 ± 10%


The poor functional recovery after Fn-n with extirpation of the ipsilateral sensory IOn was worsened by manual stimulation (massage; Fn-n + IOn-ext +Mstim) of the whisker pad (Table 1). In contrast with our earlier results, where Mstim with intact sensory supply promoted recovery of the amplitude to 51 ± 19° [62] after transection of the facial nerve, identical Mstim yielded worse recovery (amplitude of only 14 ± 6°; *p* ˂ 0.05) after facial nerve transection, plus ablation of sensory input to the paralyzed whisker pad musculature. 

### 2.2. Muscle Innervation 

The poor recovery after Fn-n was associated with the incidence of high levels of polyinnervation; namely, the innervation of each endplate by several axons or axonal branches that is normally seen only in the neonate [63,64,65]. The changed pattern of innervation was studied in the extrinsic vibrissal muscle, the levator labii superioris muscle (LLS). LLS is the largest vibrissal muscle and its permanent anatomical location, identification and dissection are relatively easy and reliable. This, in turn, renders the results after its tangential sectioning and histological staining, steady. Finally, like the numerous and very small intrinsic vibrissal muscles (actually muscle slings), LLS is supplied by six longitudinal branches of the ramus buccalis of Fn [66,67,68,69,70,71,72,73] (Figure 3).

The post-transectional polyinnervation of the motor endplate in the adult LLS contrasts with the normally monoinnervated endplates in adult muscles, where each endplate is contacted by only one axon or axonal branch. The percentage of polyneuronally innervated endplates in the reinnervated LLS was 53 ± 10% (Fn-n with intact sensory input), and 43 ± 9% (Fn-n with abolished ipsilateral sensory input; IOn-ipsi-ex). Our earlier results show that after Fn-n + Mstim, the percentage declined significantly to 22 ± 3%. This beneficial effect was abolished by trigeminal depletion by excision of the ipsilation IOn, with the percentage of polyinnervated motor endplates as high as 51 ± 10% after Fn-n + IOn-ipsi-ex + Mstim (Table 1).

### 2.3. Conclusions

Mstim mediated by intact sensory afferent connections is effective in reversing poor recovery and reinnervation of the whisker pads. When the sensory IOn was excised, the Mstim was less effective.

## 3. Mild Ipsilateral to Fn-n Trigeminal Indirect Stimulation (by Removing the Contralateral Vibrissal Hairs) and Direct Trigeminal Stimulation (by Massaging the Ipsilateral Whisker Pad) after Double Anastomotic Surgery on the Sensory Infraorbital and Motor Facial Nerves Improves the Quality of Muscle Reinnervation and Vibrissal Motor Performance. See Flow Chart Table 3 for a Synopsis

### 3.1. Experimental Rat Groups

All four groups of rats were subjected to unilateral transection and suture, i.e., anastomosis) of the facial (Fn-n) and infraorbital (IOn-n) nerves on the right side of the nose. Rats in the control group 1 received no postoperative therapy (Fn-n + IOn-n). In rats in the experimental group 2, the contralateral left vibrissal hairs were removed to ensure maximal use of the reinnervated ipsilateral ones (vibrissal stimulation; Fn-n + IOn-n + Vstim). In rats in the experimental group 3, the ipsilateral reinnervated right whisker pads were manually stimulated (massage; Fn-n + IOn-n + Mstim). In rats from the experimental group 4, the Vstim of the reinnervated vibrissae was followed by manual stimulation of the same vibrissae (Fn-n + IOn-nN + Vstim + Mstim; Figure 4; Table 3).
ijms-23-15101-t003_Table 3Table 3Experimental design flow chart depicting animal grouping and sequence of the main procedures in the section describing the effect of mild ipsilateral to Fn-n trigeminal indirect stimulation and direct trigeminal stimulation after double anastomotic surgery on the sensory infraorbital and motor facial nerves.. Fn-n, transection and suture of the facial nerve; Ion-ipsi-ex, excision of the ipsilateral infraorbital nerve; Mstim, manual stimulation; Fn-n + IOn-n + Vstim, facial nerve transection and suture + infraorbital nerve transection and suture + vibrissal stimulation.Experiment’s Research Goals Day 0 (Beginning)Day 1: Surgeries (1 Day Later)Day 2–111 Day 64–111 Day 112Day 113–118 Results**Section 3**: To determine the effect of mild trigeminal indirect stimulation of sensory nerves (by removing the contralateral vibrissal hairs and massaging the ipsilateral whisker pad), after double anastomotic surgery on the sensory infraorbital and motor facial nerve, on the quality of muscle reinnervation and vibrissal motor performance. Clipping of all vibrissal hairs except 2 on each side of the face in preparation for the videotaping.Pre-operative videotaping of 48 intact rats for video-based motion analysis of the vibrissae movements. Right Fn-n + IOn-n) in 48 rats. 12 rats (group 1) received no other treatment. In another 12 rats (group 2) the contralateral vibrissal hairs were removed (vibrissal stimulation; Fn-n + IOn-n + Vstim).In rats from group 3, the ipsilateral whisker pads were manually stimulated (Fn-n + IOn-n + Mstim).In rats from group 4, the Vstim of the reinnervated vibrissae was followed by Mstim (Fn-n + IOn-nN + Vstim + Mstim).Daily manual stimulation of whisker pads in group 3. Daily manual stimulation of whisker pads in group 4. Clipping of all vibrissal hairs except 2 on each side of the face in preparation for videotaping. Postoperative videotaping of all rats for video-based motion analysis of the vibrissae movements.Injection of 1% Fast Blue to back-label motoneuronal perikarya. Perfusion fixation with 4% PFA in 0.1 M PBS, pH 7.4 Cutting of the brainstems on a vibratome and determining the intensity of synaptophysin fluorescence. Cutting of the ipsilateral to Fn-n LLS on a cryostat and determining the degree of polyinnervation of the motor endplates.Amplitude Intact: 62 ± 6°; Fn-n + IOn-n: 11 ± 4°; Fn-n + IOn-n + Vstim: 28 ± 9°; Fn-n + IOn-n + Mstim: 30 ± 11°; Fn-n + Ion-n + Vstim +Mstim: 32 ± 10°.Polyinnervation %: Intact: 0%; Fn-n + IOn-n: 58 ± 8%; Fn-n + IOn-n + Vstim: 40 ± 3%; Fn-n + IOn-n + Mstim: 40 ± 2%; Fn-n + IOn-n + Vstim + Mstim: 33 ± 10%. Synaptophysin covered area:: Intact: 17 ± 2%; Fn-n + IOn-n: 12 ± 1%; Fn-n + IOn-n + Vstim: 13 ± 2% Fn-n + IOn-n + Mstim: 13 ± 2%; Fn-n + IOn-n + Vstim + Mstim: 12 ± 2%.


### 3.2. Functional Recovery 

Four months after the transection and anastomosis of Fn and IOn (Fn-n + IOn-n) without treatment, in the rats in control group 1 recovery was poor; the amplitude and angular velocity during whisking were reduced to about 20% of the value in intact rats (11 ± 4°). All three postoperative stimulation procedures in the experimental groups 2–4 improved function, with significantly higher amplitudes of vibrissal whisking than the control rats in group 1: Fn-n + IOn-n + Vstim 28 ± 9°; Fn-n + IOn-n + Mstim 30 ± 11°; Fn-n + IOn-n + Vstim + Mstim 32 ± 10°. Whilst the amplitudes in all the experimental groups were not significantly different, but significantly better than those in the control group 1, the clear functional improvements in vibrissal whisking amplitude were ~50% less than the values in intact animals.

### 3.3. Morphological Estimates in the Facial Muscles

Qualitative examination of LLS showed that the rats in the experimental groups 2–4 receiving Vstim, Mstim or Vstim + Mstim had larger muscle fiber diameters and a higher incidence of intramuscular axonal branches compared with rats in the control group 1, which did not receive any treatment.

These observations were matched by the quantitative assessments of the degree of polyinnervation of the endplates in the control rats with the double lesion of Fn and IOn, showing a significant decline in the percentage of polyneuronal innervation from 58 ± 8% in the control group 1 rats with no treatment to 40 ± 3%; 40 ± 4%; and 33 ± 10% for Fn-n + IOn-n + Vstim; Fn-n + IOn-n + Mstim; Fn-n + IOn-n + Vstim + Mstim in the experimental groups 2–4, respectively.

### 3.4. Morphological Estimates in the Facial Nucleus 

To provide evidence for any direct impact of trigeminal afferents on the facial motoneurons, we identified their perisomatic synapses by immunofluorescent staining with anti-synaptophysin on the perikarya of the motoneurons which had been labelled retrogradely with Fast Blue (FB) beforehand (Figure 5 and Figure 6). 

The fractional area covered by synaptophysin-positive particles in the intact lateral facial subnucleus was 17.29% ± 2.0%. The synaptophysin-positive fractional area decreased significantly in the experimental groups 2–4 and was not restored regardless of the treatment: Fn-n + IOn-n: 12 ± 1%; Fn-n + IOn-n + Vstim: 12 ± 1%; Fn-n + IOn-n + Mstim: 13 ± 2%; Fn-n + IOn-n + Vstim + Mstim: 12 ± 2%. 

There were 22.41 ± 8.54 synaptic profiles on the surface of intact facial motoneurons in the lateral facial subnucleus that were identified by retrograde labeling with FB. The corresponding numbers on the motoneurons of the rats in the experimental groups 2–4 were significantly lower, regardless of the treatment: Fn-n + IOn-n: 16 ± 7; Fn-n + IOn-n + Vstim: 16 ± 7; Fn-n + IOn-n + Mstim: 17 ± 6; Fn-n + IOn-n + Vstim + Mstim: 16 ± 7. These were also not restored.

The linear synaptic density per millimeter length of intact motoneuronal perikarya, a measure which corrects for perikaryal size, was 164.90 ± 33.00. Linear synaptic density was reduced in the experimental groups; Fn-n + IOn-n: 129 ± 35; Fn-n + IOn-n + Vstim: 126 ± 36; Fn-n + IOn-n + Mstim: 134 ± 41; Fn-n + IOn-n + Vstim + Mstim: 129 ± 36.

### 3.5. Conclusions

After transection and surgical repair of both the facial (motor) and the infraorbital (sensory) nerves, noninvasive interventions, which ensured a forced use of the vibrissae ipsilateral to injury (vibrissal stimulation, VStim, or manual stimulation (Mstim) of the whisker pads, improved recovery in whisking when compared with no treatment. The experimental methods, used to determine the overall ‘amount’ of synapses in the facial nucleus and the synaptic density on the surface of facial perikaryal, did not reveal any restoration of these synapses after reinnervation and Vstim and/or Mstim; however, it is possible that the efficacy of the synapses was increased to account for the improvement in whisking amplitude.

### 3.6. Clinical Application 

Our findings that, after facial nerve injury and surgical repair, the regenerating sensory afferents are important for recovery of motor functions, indicate the feasibility of sensory stimulation for the treatment of human patients. Our results show that sensory stimulation alone (Sstim) or prior to manual stimulation (Sstim + Mstim) after the combined trigeminal plus facial nerve lesion, is not superior to manual stimulation (Mstim) alone for functional motor recovery. This at first sight negative result has one advantage for clinical practice; namely, that the elaboration of methods of sensory stimulation, such as moist heat, of paralyzed human mimic muscles may not be necessary in physiotherapy. Easing soreness, providing motion, and increasing blood circulation, the massage (Mstim) is also good for stimulating the trigeminal afferent nerves. This method of manual/sensory stimulation is also beneficial in patients with double (facial and trigeminal) nerve lesions. Indeed, using the opposite thumb on the inside of the cheek and the 2 and 3 digits on the facial skin, patients are usually taught to draw the tissues toward the mouth. Usually, most of them report increased comfort and mobility after several weeks of practice [78]. 

## 4. Intensive Ipsilateral to Fn-n Trigeminal Indirect Stimulation, Produced by Its Forced Overuse Due to Excision of the Contralateral Infraorbital Nerve (Ion) after Surgery on the Buccal Branch of the Facial Nerve, Attenuates the Degree of Collateral Axonal Branching. See Flow Chart Table 4 for a Synopsis

### 4.1. Introduction 

It is well known that misdirection of regenerating axons may occur in three ways. *The first* is that axons are misrouted into endoneural tubes within fascicles that lead to inappropriate muscles [79,80,81]. *The second* is that, in contrast with the precise target-directed pathfinding of single motor neurites during embryonic development [82], several branches rather than one branch grow out from each transected nerve fiber [79,83,84,85,86]. *The third* means of misdirection is accomplished by the intramuscular or terminal sprouting of axons within the reinnervated target muscle [87]. 

We performed combined facial and trigeminal surgery to determine whether altered trigeminal sensory input, via ipsi- or contralateral infraorbital nerves (IOn), to the axotomized electrophysiologically silent facial motoneurons improves specificity of reinnervation. The rationale for this approach was derived from existence of direct ipsilateral and “crossed” connections between the trigeminal and facial nuclei [45,46,49,50]. 

*Methods: surgeries and rat groups.* The motor buccal nerve (Bn, or, more correctly, buccal branch of the facial nerve, Fn) and the sensory infraorbital nerve (IOn) were each transected and sutured end-to-end (anastomosis) to examine the appropriate and inappropriate regeneration of the Bn axons into its two distal stumps, the superior and inferior buccolabial nerves (ramus buccolabialis superior and inferior). 

Rats in group 1 served as unoperated controls. The experimental groups 2–4 were used for comparative assessment of nerve regeneration and branching. Rats in the experimental groups were subjected to identical transection and suture of the right Bn (buccal–buccal anastomosis, Bn-n). The rats of Groups 2–4 had the buccal nerve transected and surgically repaired, but the rats in group 3 underwent Bn-n plus excision of the ipsilateral (right) IOn (Bn-n + IOn-ipsi-ex), and those of Group 4 underwent Bn-n plus excision of the contralateral (left) IOn (Bn-n + IOn-contra-ex; Table 4).
ijms-23-15101-t004_Table 4Table 4Experimental design flow chart depicting animal grouping and sequence of the main procedures in the section describing the effect of intensive trigeminal indirect stimulation, produced by its forced overuse due to excision of the contralateral infraorbital nerve (Ion) after surgery, on the buccal branch of the facial nerve on the degree of collateral axonal branching. Bn-n, transection and suture of the buccal branch of the facial nerve.Experimental SetResearch GoalsDay 1: Surgeries Day 28 (4 Weeks Later)Day 38 (10 Days Later) Day 38Results**Section 4**: Effects of the intensive trigeminal indirect stimulation (excision of the contralateral infraorbital nerve) after surgery on the buccal branch of the facial nerve, on the degree of collateral axonal branching at the lesion site.All groups of rats consisted of 6 animals. Rats in group 1 served as unoperated controls. Rats in groups 2–4 were subjected to identical transection and suture of the right buccal branch of the facial nerve (buccal–buccal anastomosis, Bn-n). The rats of group 3 underwent Bn-n plus excision of the ipsilateral (right) IOn (Bn-n + IOn-ipsi-ex), and those of group 4 underwent Bn-n plus excision of the contralateral (left) IOn (Bn-n + IOn-contra-ex).Retrograde labeling of the facial motoneurons with fluorescent tracers, Fluoro-Gold (FG; Fluorochrome Inc., Englewood, CO, USA) and DiI (Molecular Probes, The Netherlands), applied to the superior and inferior buccolabial nerves (BLn) respectively.Perfusion fixation with 4% paraformaldehyde in phosphate-buffered saline, pH 7.4 Cutting of the brainstems on a vibratome and determining the number of double-labelled (FG + DiI) perikarya which indicates the degree of collateral axonal branching at the site of nerve injury. Portion of double-labelled perikarya: Intact: 0%; Bn-n: 23%; Bn-n + IOn-ipsi-ex: 13%; Bn-n + IOn-contra-ex: 8% 

### 4.2. Identification and Localization of Facial Motoneurons Regenerating into the Branches of the Buccal Nerve after Bn-n Surgery 

Retrograde labeling of the facial motoneurons (Figure 7A,B) was performed with fluorescent tracers, Fluoro-Gold (FG; Fluorochrome Inc., Englewood, CO 80155, USA) and DiI (1,1′-dioctadecyl-3,3,3′,3′-tetramethylindo-carbocyanine perchlorate, Molecular Probes, The Netherlands), applied to the superior and inferior buccolabial nerves (BLn). 

### 4.3. Results

#### 4.3.1. Rat Group 1: Distribution of Facial Motoneurons in Both Branches of the Buccal Nerve in Unoperated Rats 

All 1858 ± 424 of the FG and the DiI back-labelled motoneurons, whose nerve fibers were localized in either the superior BLn or the inferior BLn branches of the buccal nerve, were localized exclusively in the lateral facial subnucleus. The 1724 ± 375 FG-labelled neurons whose fibers were exposed to FG in the superior BLn were found in the ventrolateral portion, and the 134 ± 125 DiI-labelled neurons exposed to DiI in the inferior Bln were in the dorsomedial portion: 91% and 9% of the total, respectively. There were no double-labelled neurons (Figure 7).

#### 4.3.2. Rat Group 2: Buccal–Buccal Anastomosis (Bn-n)

All the motoneurons regenerated their axons 28 days after Bn-n, but the myotopic organization of the lateral facial subnucleus into the ventrolateral and dorsomedial areas for the superior and inferior BLn was greatly altered (Figure 7D). The percentage of FG-labelled motoneurons in the ventrolateral area that regenerated nerve fibers into the superior BLn declined from 91% to ~56%, and those of the DiI-labelled neurons in the dorsomedial area that regenerated their nerves into the inferior BLn increased from 9% to ~21%. The changes were statistically significant (*p* < 0.05). The double-labelled neurons comprised 23% of all the motoneurons in the lateral facial subnucleus that had regenerating nerve fibers into both superior and inferior BLn. The percentage fell to 11% by 112 days after Bn-n, consistent with the pruning of excess regenerated axon sprouts, as reported earlier in femoral and sciatic regenerating nerve fibers [83,88,89]. 

#### 4.3.3. Rat Group 3: Bn-n Plus Excision of the Ipsilateral Infraorbital Nerve (IOn)

Again, there was no significant change in the number of motoneurons regenerating their axons with *all* 1776 ± 476 localized in the lateral facial subnucleus. Again, the myotopical organization of the motoneurons was lost (Figure 7F), and the percentage of FG-labelled motoneurons in the ventrolateral area that regenerated nerve fibers into the superior BLn declined from 91% to ~48%; those of the DiI-labelled neurons in the dorsomedial area that regenerated their nerves into the inferior BLn increased from 9% to ~39%, with 13% of the motoneurons regenerating their nerve fibers into both BLn branches.

#### 4.3.4. Rat Group 4: Bn-n Plus Excision of the Contralateral Infraorbital Nerve (IOn)

The number of 1855 ± 581 motoneurons regenerating their nerve fibers was the same as those of intact motoneurons in the lateral facial subnucleus. Importantly, the 56% and 48% of these FG back-labelled motoneurons in the dorsomedial area of the subnucleus that regenerated their nerve fibers into the appropriate superior BLn branch, after Bn-n alone and after Bn-n and excision of the ipsilateral IOn, respectively, increased significantly to 69% after Bn-n, when the *contralateral* IOn nerve was excised. The mean number of neurons whose axons projected into the inferior buccolabial nerve decreased to 23%, and the mean number of double-labeled neurons decreased to about 8% of all motoneurons in the lateral facial subnucleus.

**Figure 7 ijms-23-15101-f007:**
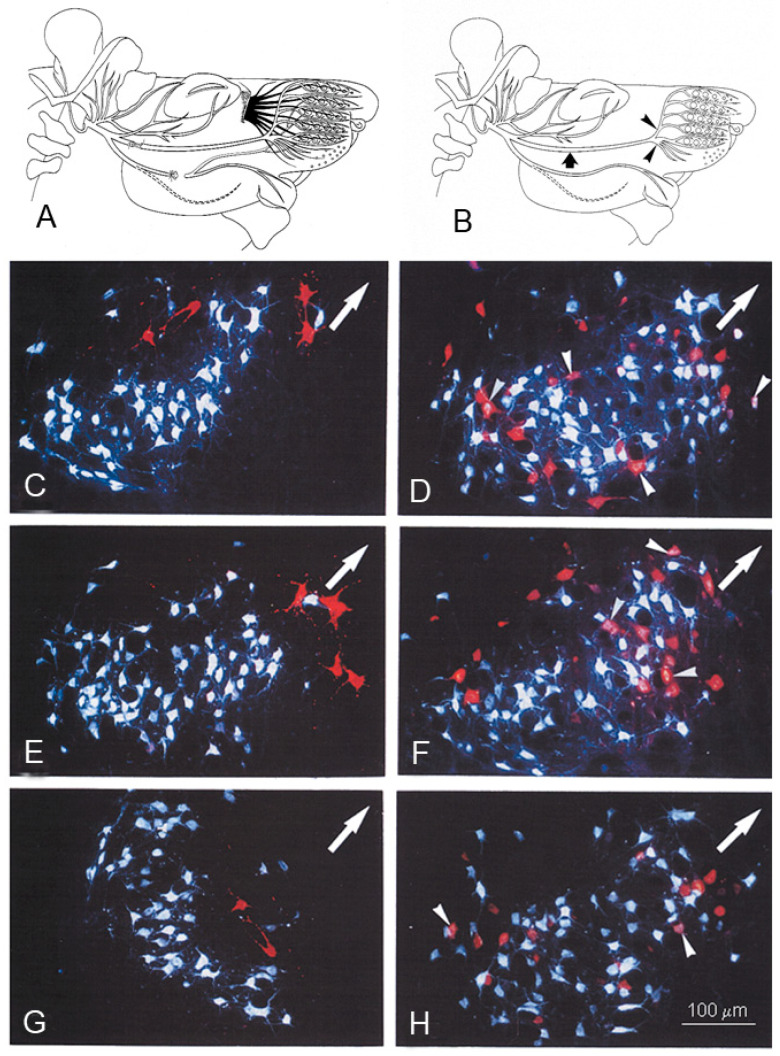
(**A**): Schematic drawing illustrating the transection and suture site of the buccal branch of the facial nerve (Bn-n) and the resection of Ramus marginalis mandibulae of the rat facial nerve. (**B**): The *large arrow* indicates the site of transection of the buccal branch. *Arrowheads* point at the sites of tracer application in the superior and inferior buccolabial nerves. (**C**–**H**): The rat brainstem 28 days after Bn-n. The photographs were produced by double exposure. The dorsomedial portion of the facial nucleus is indicated by an *arrow*. (**C**): The contralateral unlesioned lateral facial subnucleus with myotopic organization of the motoneurons, whose axons project into the superior buccolabial nerve [retrogradely labelled in *white* by Fluoro-Gold (FG)] and into the inferior buccolabial nerve (labelled in *red* by DiI). Most FG-labelled motoneurons are localized in the ventrolateral portion, and those labelled with DiI are in the dorsomedial part of the subnucleus. (**D**): The lesioned lateral facial subnucleus after Bn-n and application of FG to the superior, and DiI to the inferior buccolabial nerves. Note the complete lack of myotopic organization with the FG-labelled (*white*), DiI-labelled (*red*), and DiI + FG-labelled (*arrowheads*) motoneurons scattered throughout the entire lateral facial subnucleus. (**E**): The rat brainstem 28 days after Bn-n *and* excision of the ipsilateral infraorbital nerve with the contralateral unlesioned lateral facial subnucleus having myotopic organization. **(F**): The lesioned facial subnucleus 28 days after Bn-n *and* excision of the ipsilateral infraorbital nerve. (**G**): The rat brainstem 28 days after Bn-n *and* excision of the contralateral infraorbital nerve. The contralateral unlesioned lateral facial subnucleus has the normal myotopic distribution of the motoneurons. (**H**): The lateral facial subnucleus 28 days after Bn-n *and* excision of the contralateral infraorbital nerve; 50 μm vibratome sections. From Angelov et al. (1999) [90].

### 4.4. Conclusions

In summary, excision of the contralateral but not the ipsilateral infraorbital nerve reduced the collateral branching, and/or increased elimination of the branches of regenerating nerve fibers after buccal nerve transection and surgical repair. This beneficial effect diminishes the extent of the inevitable postoperative misdirected reinnervation of target muscles by regenerating nerve fibers. A possible explanation for this effect of the contralateral sensory injury is that the excision of IOn forces the rat to use the side with the intact sensory innervation. This, however, is the same side as that with the transected facial axons. Thus, we may assume that the intact ipsilateral afferent input strongly stimulates the axotomized motoneurons to elongate axons, to innervate the piloerector muscles and thus to provide motor basis for vibrissae rhythmical whisking.

## 5. Intensive Ipsilateral to Fn-n Trigeminal Indirect Stimulation, Produced by Its Forced Overuse Due to Excision of the Contralateral Infraorbital Nerve (Ion) after Surgery on the Buccal Branch of the Facial Nerve, Improves the Accuracy of Muscle Reinnervation. See Flow Chart Table 5 for a Synopsis 

### 5.1. Research Question 

In order to determine whether the significantly reduced misdirection of the regenerating buccal motor nerve axons that occurred when the contralateral but not the ipsilateral IOn was excised, we carried out the same surgeries as in the three experimental rat groups used in the last section, to ask whether the removal of the sensory input from the ipsi (right)- or contralateral (left) vibrissae to the right axotomized buccal motoneurons would attenuate the misguidance of regrowing buccal nerve sprouts, and thereby improve the accuracy of target muscle reinnervation (Table 5).
ijms-23-15101-t005_Table 5Table 5Experimental design flow chart depicting animal grouping and sequence of the main procedures in the section, describing the effect of intensive trigeminal indirect stimulation, produced by its forced overuse due to excision of the contralateral infraorbital nerve (Ion) after transection and suture of the buccal branch of the facial nerve (Bn-n), on the accuracy of muscle reinnervation.Experimental Set Research Goals 10 Days before SurgeryDay 0: SurgeriesDay 28(4 Weeks Later)Day 38 (10 Days Later) Day 38Results **Section 5**: Effect of intensive trigeminal indirect stimulation (excision of the contralateral IOn) after surgery on Bn on the accuracy of muscle reinnervation.Buccal motoneurons were back labelled prior to surgeries in all 4 rat groups (each of 6 rats) to identify and localize their motoneuronal perikarya in the brainstem. A solution of 1% Fluoro-Gold (FG) was injected into the muscles of the whisker pad. Rats in group 1 served as unoperated controls. Rats in groups 2–4 were subjected to identical transection and suture of the right buccal branch of the facial nerve (buccal–buccal anastomosis, Bn-n). The rats of group 3 underwent Bn-n plus excision of the ipsilateral (right) IOn (Bn-n + IOn-ipsi-ex), and those of group 4 underwent Bn-n plus excision of the contralateral (left) IOn (Bn-n + IOn-contra-ex).28 days after surgery, a 1% solution of Fast Blue (FB) was injected into the same muscle site as the earlier FG injection.Perfusion fixation with 4% paraformaldehyde in phosphate-buffered saline, pH 7.4Cutting of the brainstems on a vibratome and determining the number of double-labelled (FG + DiI) perikarya which indicates accuracy of muscle reinnervation. Portion of double-labelled perikarya: Intact: ~90–100%; Bn-n: 27%; Bn-n + IOn-ipsi-ex: 32%; Bn-n + IOn-contra-ex: 41% 


### 5.2. Surgeries and Rat Groups 

As in the experiments described in Section 6, the rat control group 1 served as unoperated control. The experimental groups 2–4 were used for comparative assessment of nerve regeneration and branching, with all the rats in the experimental groups subjected to identical transection and suture of the right Bn (buccal–buccal anastomosis, Bn-n); however, the rats of Group 3 underwent Bn-n plus excision of the ipsilateral (right) IOn (Bn-n + IOn-ipsi-ex), and those of Group 4 underwent Bn-n plus excision of the contralateral (left) IOn (Bn-n + IOn-contra-ex).

### 5.3. Identification and Localization of Buccal Motoneurons Regenerating into Muscles of the Whisker Pad

Buccal motoneurons were back labelled prior to surgeries in all four rat groups to identify and localize their motoneuronal perikarya in the pons. A solution of 1% Fluoro-Gold (FG) was injected into the muscles of the whisker pad prior to surgeries in all the rat groups and, 28 days later, a 1% solution of Fast Blue (FB) was injected into the same muscle site as the earlier FG injection. The midbrains of the rats were removed to cut transverse sections for identification of the back-labelled facial motoneurons. With custom-made selective filter sets for FG and FB, separate digital images of motoneuronal profiles, orange-red and blue, respectively, were saved for analysis of location and number.

### 5.4. Results

#### 5.4.1. Rat Group 1

In the control group of unoperated rats, the percentage of double-labelled facial motoneurons that were back labelled with FG, and then FB-injected into the same site within the muscles of the whisker pad, was ~90%; this is reasonably close to the theoretical expectation of 100% double-labeling (Figure 8A–C). They were all located in the lateral facial subnucleus and their numbers were in the normal range (Figure 8G,J).

#### 5.4.2. Rat Group 2: Bn-n

Only 398 ± 80 facial motoneurons (27%), were double labelled by FG and FB dyes, demonstrating that the remaining 73% neurons labelled only with FG had *not* regenerated into the same muscles in the whisker pad where FG was injected prior to the Bn-n surgery. These latter “ectopic” motoneurons were located in the dorsal and intermediate facial subnucleus (Figure 8D–F).

Reinnervation of the whisker pad by “ectopic” motoneurons in the dorsal, intermediate, medial, and ventromedial facial subnuclei has been previously reported in our earlier studies after transection and suture of the facial (but not buccal) nerve [92,93,94]. The motoneurons in these subnuclei normally extend their axons into the zygomatic, marginal mandibular, posterior auricular and cervical branches of the facial nerve [76], which remained intact in these studies, i.e., they should not grow axonal branches. We have no explanation for the occurrence of axonal branches from intact nerves. Perhaps the trauma caused by Bn-n surgery was a strong enough insult to trigger axonal branching from the adjacent intact nerves. 

#### 5.4.3. Rat Group 3: Bn-n Plus Excision of the Ipsilateral IOn (Bn-n + IOn-ipsi-ex) 

Four weeks after the surgery, 436 ± 68 facial motoneurons were double labelled (pink-purple), that is, 32% of the motoneurons which reinnervated the whisker pad post-surgery were in the original midbrain pool of motoneurons, the number not being statistically different from that after Bn-n surgery without excision of the IOn-ipsi. Postoperatively labelled facial motoneurons were located “correctly” in the lateral facial subnucleus and “incorrectly” in the medial, and intermediate facial subnuclei (Figure 8H,I).

#### 5.4.4. Rat Group 4: Bn-n Plus Excision of the Contralateral IOn (Group BBA + ION-Contra-ex)

Four weeks after surgery, there were 580 ± 63 double-labelled motoneurons, i.e., ~41% of the motoneurons, which innervated the whisker pad muscles before surgery, sent an axon or an axonal branch to their original target muscle in the whisker pad (Figure 8L), a statistically significant increase in the number of double-labelled motoneurons as compared with the numbers after Bn-n + IOn-ipsi-ex (*p* = 0.01). In addition, the motoneurons appeared larger than the shrunken perikarya observed in the previous groups. Postoperatively labelled motoneuronal somata with FB were also found in the intermediate facial subnucleus (Figure 8K).

### 5.5. Conclusions

The data from the experiments, in which motoneurons in the lateral facial subnucleus were back labelled by applying fluorescent dyes to the target muscles of the whisker pad, show that when compared to transection and suture alone (Bn-n), the lesion of the contralateral infraorbital sensory nerve improved the accuracy of target muscle reinnervation. 

### 5.6. Discussion 

These morphological results correlate with preliminary observations that recovery of vibrissal motor performance started in the group Bn-n + IOn-contra-ex earlier than that in the groups Bn-n and Bn-n + IOn-ipsi-ex. These observations were not mentioned in the results because they were not as plausible and persuasive as data from video-based motion analysis. Nonetheless, our observations are consistent with the beneficial behavioral impacts of various combined lesions on the facial nerve in rats observed previously (see Huston et al. 1990 for a review [95]), and with some known benefits of physical therapy in human patients [96,97,98]. Unfortunately, the mechanisms of such improvements are still unknown. 

We offer two speculative theories to explain why a contralateral trigeminal sensory nerve lesion (excision of the infraorbital nerve, IOn) improves the accuracy of muscle target reinnervation by regenerating facial motoneurons. *The first theory* assumes that the observed slower regrowth and more abundant axonal branching on the side of the face ipsilateral to the Bn-n and IOn-excision surgery are driven by the abnormal excitability with hyperstimulation of the spinal trigeminal nucleus, *because* of the extirpation of the sensory IOn. This unnatural, injury-triggered excitability may result from (1) the proximal stump developing a neuroma, as a source of sustained long term ectopic impulses [37]; (2) 15–20% of the injured sensory neurons and their fibers degenerating with preferential loss of the small caliber fibers [99,100] and the larger Aδ-fibers occupying the “vacant room” of the degenerated Aβ- and C-fibers [101]; and (3) reduced inhibitory control in the corresponding spinal trigeminal nucleus, resulting from the selective degeneration of GABAergic neurons [40]. This abnormal activity could affect the motoneurons in the lateral facial subnucleus via the stronger ipsilateral trigemino-facial projection. Thereby, the motor fibers emanating from those motoneurons may release an excess of glutamate, which is known to suppress neurite outgrowth [102]. Alternatively, because the crossed trigemino-facial projection is “weaker” than the ipsilateral one [48], the neurotoxic effect of glutamate may not be sufficient to cause glutamate-mediated damage. There is also evidence that GABA enhances neurite outgrowth [103] such that GABAergic interneurons may be involved [104]. In summary, the first theory assumes that the slower regrowth and the more abundant branching of axons on the side ipsilateral to the combined (facial and trigeminal) surgery are due to hyperstimulation in the ipsilateral trigeminal nucleus. 

*The second theory* assumes that the ipsilateral to Bn-n excision of IOn deprives the ipsilateral facial motoneurons of trigeminal sensory input, and “forces” the rat to use the other side of the face, which is with intact sensory innervation. Even after Bn-n only, rats preferentially use the contralateral side, which is with preserved sensitivity, for sensory input. This deprives the lesioned facial motoneurons of trophic inputs initiated by the powerful ipsilateral sensory afferents. The unilateral transection of both the facial and Ion creates an even greater advantage for the intact side, at the expense of the lesioned side. Consequently, regeneration proceeds more slowly and with abundant axonal branching. 

In contrast, when the IOn is excised contralateral to the Bn-n, the rat preferentially uses the side with the intact sensory innervation. This, however, is the same side as that with the transected facial axons. Thus, it might be that the intact ipsilateral afferent input strongly stimulates the axotomized motoneurons to elongate axons, to innervate the piloerector muscles and thus to provide motor basis for vibrissae rhythmical whisking. 

### 5.7. Clinical Implications 

The method of combined transection of both the facial and trigeminal nerves has no application in human patients. Still, the surprisingly good morphological results after excision of the contralateral IOn suggest that during the post-facial surgery period, a temporary blockade of the trigeminal fascicles, by paraneural injections of various anaesthetics, for example, might yield similar beneficial results in humans. We anticipate such a clinical experiment. 

From clinical experience, after reconstructive surgery on the facial nerve, it is well known that conditioning rehabilitation improves the speed and specificity of reinnervation. This has been supported by the rare occurrence of synkinesia in patients who undergo conditioning rehabilitation [91,105]. Activation of the low threshold mechanoreceptors in the corresponding region of the face has been frequently used as a conditioning stimulus. Accordingly, increased sensory input may exert a two-fold effect on the regenerating facial motoneurons of (1) speeding axonal elongation and (2) decreasing aberrant innervation. Our experimental data fit well with this possibility. The rat facial nucleus receives a heavy, mainly ipsilateral projection from second-order somatosensory neurons situated throughout the spinal trigeminal complex. Yet, despite our expectation that the ipsilateral trigeminal nerve would play the most important role in the sensory-motor integration at the level of the facial nucleus, we found significantly more accurate regeneration if Bn-n was accompanied by a lesion of the contralateral trigeminal nerve. 

## 6. Direct Stimulation of the Trigeminal Nerve Afferents (by Manual Stimulation of the Whisker Pad, Massage) after Facial Nerve Surgery Restores the Synaptic Density in the Facial Nucleus, Improves the Quality of Target Reinnervation and Promotes Recovery of Vibrissal Motor Performance. See Flow Chart Table 6 for a Synopsis

### 6.1. Experimental Rationale 

After complete spinal cord transection, motoneurons distal to the injury undergo atrophy; their dendritic trees shrink and become partially deafferented [106,107]. Similarly, axotomy causes deafferentation of motoneurons and changes in their somata and dendritic trees [19,22]. 

### 6.2. Methods 

*Rat groups. The rat control group 1* comprised unoperated rats. The facial nerve was cut and re-sutured (Fn-n) in the experimental groups 2 and 3. In the former group, Fn-n + handling, the rats were held in the experimenter’s hand, not receiving any stimulation of the vibrissal muscles. In the latter Group 3, Fn-n was followed by manual stimulation of the whisker pad (Fn-n + Mstim; Table 6). 

**Table 6 ijms-23-15101-t006:** Experimental design flow chart depicting animal grouping and sequence of the main procedures in the section describing the effect of direct stimulation of the trigeminal nerve afferents after facial nerve surgery (Fn-n) on the synaptic density in the facial nucleus, the quality of target reinnervation and recovery of vibrissal motor performance. Mstim, manual stimulation.

Experiment’s Research Goal	Day 0 (Beginning)	Day 1: Surgeries (1 Day Later)	Day 2–56 (2 Months)	Day 57	Day 58–67	Results
**Section 6**: Effect of direct stimulation of the trigeminal nerve afferents (by manual stimulation of the whisker pad, massage) after facial nerve surgery on the synaptic density in the facial nucleus.	Clipping of all vibrissal hairs except 2 on each side of the face in preparation for the videotaping. Pre-operative videotaping of all intact rats for video-based motion analysis of the vibrissae movements.	6 animals in group 1 were intact controls. The facial nerve was cut and re-sutured (Fn-n) in the experimental groups 2 and 3. In group 2 (6 rats; Fn-n + handling), the rats were held in the experimenter’s hand, not receiving any stimulation of the vibrissal muscles. In group 3 (r rats), Fn-n was followed by manual stimulation of the whisker pad (Fn-n + Mstim).	Daily manual stimulation of whisker pads in group 3.	Clipping of all vibrissal hairs except 2 on each side of the face in preparation for videotaping. Postoperative videotaping of all rats for video-based motion analysis of the vibrissae movements.	Perfusion fixation with 4% PFA in 0.1 M PBS, pH 7.4 Cutting of the brainstems on a vibratome and determining the intensity of synaptophysin fluorescence Cutting of the ipsilateral to Fn-n levator labii superioris muscle (LLS) on a cryostat and determining the degree of polyinnervation of the motor endplates.	Amplitude Intact: 57 ± 13° Fn-n + handl: 19 ± 6°; Fn-n + Mstim: 51 ± 19°; Polyinnervation %: Intact: 0% Fn-n + handl: 53 ± 10%; Fn-n + Mstim: 22 ± 5%; Amount of synaptophysin-positive terminals in the facial nucleus: Intact: 34.3 × 10^6^ ± 2.3 × 10^6^; Fn-n + handl: 29.2 × 10^6^ ± 1.8 × 10^6^; Fn-n + Mstim: 33 × 10^6^ ± 2.6 × 10

*Video-based motion analysis* was used to study vibrissal motor performance during explorative whisking. 

*Motor endplate reinnervation*. Two months after surgery, combined immunocytochemistry with class III-tubulin antibody and histochemistry with α-bungarotoxin was used to determine the pattern of motor endplate reinnervation in the LLS muscle. 

*Synaptic input on facial motoneurons*. Synaptophysin immunocytochemistry was used to quantify the extent of total synaptic input to motoneurons in the facial nucleus, synaptophysin being an established pan marker for presynaptic terminals. Images were obtained on an epi-fluorescence microscope from sections stained with a highly diluted (1:4000) anti-synaptophysin antibody for 2 h. This protocol allowed us to obtain photo images in which, comparable to thin confocal optical sections, numerous puncta within the neuropil and around motoneuronal cell bodies in the facial nucleus were clearly discernible (Figure 9). To quantitate synaptophysin levels in the facial nucleus, we used a grayscale-based densitometric approach [108,109] for analyses of the frequency distributions of pixel intensities. Fluorescent images were compared using the 8 BPP gray scale format. Thereby, each pixel contains 8 bits of information encoding brightness, which ranges on a scale from 0 to 255. The scale for pixel brightness, or pixel gray value, is constructed so that the higher numbers indicate greater pixel brightness. Digital images were captured with a slow scan CCD camera (Spot RT, Diagnostic Instruments, 540 Burroughs Sterling Heights, MI 48314 USA). For quantification of pixel brightness, images were captured using the ×16 objective and the Image-Pro Plus Software Version 5.0 (Media Cybernetics, Inc., Silver Spring, MD, USA). Exposure time was optimized to ensure that only few pixels were saturated at 255 gray value. All images were taken under same conditions of exposure (i.e., duration). An interactive threshold was used to detect the pixel brightness of the minimum fluorescence. Threshold values ensured the inclusion of the entire signal range in the sample. This threshold value was further used to extract and compare the pixel number between all animals of the same group and between experimental groups. The background intensities were identical from image to image, around a pixel gray value of 30. Accordingly, the threshold level for measuring pixel number and brightness was set at 30, and the range for measuring pixel number and brightness was set at 30–129.

### 6.3. Results 

#### 6.3.1. Recovery of Whisking

*Recovery of whisking* after Fn-n + handling was poor. Mstim of the ipsilateral whisker pad for 5–10 min daily had a dramatic effect, resulting in an almost complete return of normal whisking, as indicated by the amplitude of movement as well as by the speed during protraction. 

#### 6.3.2. Loss of Synapses in the Lesioned Facial Nucleus 

Synaptic terminals rapidly detach from motoneurons after injury, a phenomenon well known as “synaptic stripping” [22,110] that accounts for the observed decline in motor nerve activity [29]. This post-traumatic deafferentation is reversible if and when target reinnervation occurs [29,111,112,113]. 

In the *experimental group 2* with Fn-n + handling, the total number of pixels was significantly lower when compared with that in control intact rats (29.2 × 10^6^ ± 1.8 × 10^6^ versus 34.3 × 10^6^ ± 2.3 × 10^6^; *p* = 0.036, ANOVA with Bonferroni post hoc test). This indicates a reduction of approximately 15%. In the experimental rat group 3 receiving Mstim after Fn-n, the mean pixel number was not statistically different from the control intact rats (33 × 10^6^ ± 2.6 × 10^6^), and was higher than in the Fn-n + handling group (*p* = 0.007). These results indicate that the synaptic input to injured facial motoneurons after Fn-n + handling is reduced, and that this loss is counteracted by Mstim. 

*Functional outcome correlates with quality of target muscle reinnervation*. Qualitative examination revealed two major features of the m. levator labii superioris. Compared with rats that received Fn-n + Mstim, the incidence of intramuscular axonal branches was higher, and diameters of muscle fibers smaller, in rats with Fn-n + handling. 

Our qualitative observations were matched by quantitative assessments of vibrissal function and the degree of polyinnervation. Thus, rats with Fn-n + handling had a high percentage of polyinnervated endplates (53 ± 10%). By contrast, rats with Fn-n + Mstim had normal vibrissal function and the degree of polyinnervation endplates was significantly smaller (22 ± 5%). 

### 6.4. Discussion 

Sensory input is an essential, yet unexploited, factor influencing motor recovery after nerve injury. In the nerve lesion paradigm used here, motor axons were lesioned, but the circuitry conveying sensory information from the facial fur to the facial motoneurons via the trigeminal nerve was intact. Our findings that normal synaptic density was restored on axotomized facial motoneurons after the surgical anastomosis of their transected nerve stumps (Fn-n), when the surgery was followed by daily manual stimulation for 5–10 min (Mstim), is consistent with the Mstim preventing the retraction of the synaptophysin-positive afferent synapses from the motoneurons and the restoration of the whisking amplitude [41]. Mstim-enhanced sensory input may also exert direct effect on the perikarya and dendrites of motoneurons, leading to enhanced production of growth-associated molecules such as growth-associated protein 43 (GAP-43), synapsin I, cAMP, and brain-derived neurotrophic factor (BDNF) which stimulate dendrite growth and synaptic remodeling [114,115]. Mstim also reduced the extent of polyneuronal muscle fiber reinnervation, aiding in the recovery of muscle function after facial nerve injury. Later, functional recovery was reported after facial nerve crush [116] but without reference to our findings on the positive effects of manual stimulation of the afferents in the whisker pad [62], and by replicating our Figure 1 in Pavlov et al. [41] without permission.

### 6.5. Mechanisms 

English and colleagues demonstrated that the retraction of excitatory glumatergic synaptic inputs onto axotomized lumbosacral motoneurons could be prevented by sex-specific exercise programs of continuous and interval training for 2 weeks [117,118]. Such retention of synaptic inputs on facial motoneurons may very well be the explanation for the positive effects of the sensory input from Mstim of the whisker pad in restoring normal whisking. 

The sensory input is likely conveyed by infraorbital nerve fibers, via the maxillary nerve to the trigeminal nucleus in the brainstem that is recognized to be essential in generating facial muscle responses and blink reflexes [119,120,121,122], and is interconnected with the subcortical central whisking pattern generator [123,124,125,126,127]. However, it is likely that the neural pathways responsible for the restoration of normal whisking behavior are more complex, considering the diversity of projections to the facial nucleus that include the neocortex, other nuclei of the cranial nerves, and the reticular formation [128,129]. Nonetheless, Mstim provides enhanced afferent input to the sensory and motor cortices, thereby maintaining appropriate levels of excitability within the trigeminal–cortical–facial loop [130,131]. In this way, intact sensitivity exerts an indirect effect on motoneurons. The notion that stimulation of intact networks improves functions in damaged circuitries is supported by follow-up results in human patients: stimulation of paralyzed facial muscles, and electromyography or neuromuscular re-education programs are very effective for increasing facial muscle control and function, even in cases of long-standing paralysis [132,133,134]. 

There may also be local positive intramuscular effects of Mstim on muscle fibers and Schwann cells that are conveyed to the axotomized motoneurons upon contact of the regenerated axons with the denervated muscle fibers. Of interest and importance in this regard is our finding that Mstim significantly reduced the degree of the polyinnervation of muscle endplates in the reinnervated levator labii superioris (LLS) muscle [73]. This is because our previous work indicates that post-lesional polyinnervation of the endplates, namely, the innervation by two or more axons, is deleterious to the recovery of facial motor function [73]. Although claimed to be transient [135], its persistence for months after nerve–muscle contact [79,89,136,137,138,139,140,141] likely limits functional recovery after nerve injury [142,143,144,145,146,147]. Muscle activity imposed artificially during reinnervation of denervated endplates reduces intramuscular sprouting [148,149,150] and, thereby, may account for Mstim reducing polyneuronal reinnervation. The link between reduced polyinnervation of the endplates *and* restoration of normal whisking provides strong evidence of the link between these sequelae of facial nerve injury and repair. 

## 7. Muscle Neurotrophic Factors Are Unlikely to Play a Role in the Reduced Polyneuronal Muscle Reinnervation by Manual Stimulation of the Denervated Whisker Pad (Mstim). See Flow Chart Table 7 for a Synopsis 

### 7.1. Experimental Rationale and Questions 

Comparisons between the expression of neurotrophic factors mRNA and protein in blind rats from the Sprague Dawley (SD)/RCS strain, which display good recovery of whisking, and SD-rats with normal vision but poor recovery of whisking function, after facial nerve transection and suture (Fn-n), revealed different time courses of increases in gene expression and protein of brain-derived neurotrophic factor (BDNF); fibroblast growth factor-2 (FGF2); and corresponding reductions in insulin growth factors 1 and 2 (IGF1, IGF2) and nerve growth factor (NGF) [151]. We asked whether analysis of some of these neurotrophic factors (FGF2, IGF1 and NGF) might reveal whether such changes in neurotrophic factor expression may be involved in the recovery of excellent functional whisking after manual muscle stimulation (Mstim; Table 7).
ijms-23-15101-t007_Table 7Table 7Experimental design flow chart depicting animal grouping and sequence of the main procedures in the section describing the role of muscle neurotrophic factors in the reduced polyneuronal muscle reinnervation by manual stimulation of the denervated whisker pad. Fn-n, transection and suture of the facial nerve. Mstim, manual stimulation.Experiment’s Research Goal Day 0 (Beginning)Day 1: Surgeries (1 Day Later)Day 2–56 (2 Months) Day 57Day 58–67 Results**Section 7**: To determine the role of muscle neurotrophic factors in the reduced polyneuronal muscle reinnervation by manual stimulation of the denervated whisker pad (Mstim)Clipping of all vibrissal hairs except 2 on each side of the face in preparation for the videotaping.Pre-operative videotaping of all intact rats for video-based motion analysis of the vibrissae movements. 6 animals of group 1 were intact controls. The facial nerve was cut and re-sutured (Fn-n) in the experimental groups 2 and 3. In group 2 (6 rats, Fn-n + handling), the rats were held in the experimenter’s hand, not receiving any stimulation of the vibrissal muscles. In group 3, (6 rats) Fn-n was followed by manual stimulation of the whisker pad (Fn-n + Mstim).Daily manual stimulation of whisker pads in group 3. Clipping of all vibrissal hairs except 2 on each side of the face in preparation for videotaping. Postoperative videotaping of all rats for video-based motion analysis of the vibrissae movements.Blood rinse by 0.1 M PBS, pH 7.4. Samples from LLS were taken and frozen. The translation of the proteins FGF2, IGF1 and NGF was determined using sandwich ELISA-Kits following the manufacturer’s instructions.FGF2 Intact: 75 ± 16 pg/mg, FGF2 Fn-n+ handling: 69 ± 24 pg/mg FGF2 Fn-n + Mstim: 56 ± 11 pg/mg IGF1 Intact: 1492 ± 87 pg/mg IGF1 Fn-n + handling 2079 ± 300 pg/mg,IGF1 Fn-n + Mstim: 1821 ± 784 pg/mg NGF Intact: 61 ± 22 pg/mg NGF Fn-n + handling: 51 ± 21 pg/mg NGF Fn-n + Mstim: 33 ± 13 pg/mg.


### 7.2. Methods 

The same rat *groups from experiments described in Section 8 were used; namely, the control Group 1 of* unoperated rats and the experimental groups 2 and 3, in which the facial nerve was cut and re-sutured (Fn-n). In the former group (Fn-n + handling), the rats were held in the experimenter’s hand, not receiving any stimulation of the vibrissal muscles. In the latter Group 3, Fn-n was followed by manual stimulation of the whisker pad (Fn-n + Mstim).

*ELISA analysis of protein content in the reinnervated levator labii* superioris (LLS) muscle; LLS muscle samples were taken and frozen 8 weeks after Fn-n and the two experimental treatments. The frozen LLS samples (30 mg) were crushed using a pebble mill with additional ultrasound treatment on ice, in a lysis buffer made of complete Mini (Roche, 68305 Mannheim, Germany; Cat. No.11836153001), HALT Protease Inhibitor Cocktail (Thermo Scientific, 28199 Bremen, Germany, Cat No. 78410) and PMSF (Applichem, 64291 Darmstadt, Germany, Cat. No. A0999), according to the manufacturer’s instructions. After centrifugation, the total protein content was measured in the supernatant using a Bradford–Assay (Serva, 69115 Heidelberg, Germany, CatNo. 39222). The translation of the proteins FGF2, IGF1 and NGF was determined for each sample, in duplicate, using sandwich ELISA-Kits (Wuhan Fine Biotech, DLDevelop; Hubei, 430206 China), following the manufacturer’s instructions. All color measurements were performed in a FLUOstar Omega (BMG Labtech, 77799 Ortenberg, Germany).

### 7.3. Results 

No differences in the protein levels of any of the neurotrophic factors were detected between rats subjected to Fn-n + handling (group of 6 rats) and Fn-n + Mstim (group of 6 rats). Mean total protein levels of 75 ± 16 pg/mg, 69 ± 24 pg/mg and 56 ± 11 pg/mg FGF2 were measured in the LLS of intact rats, 2 months after Fn-n + handling, and Fn-n + Mstim, respectively. In the same number of rats, the amount of IGF1-protein in the intact LLS was 1492 ± 87 pg/mg total protein, and 2 months after Fn-n, we measured 2079 ± 300 pg/mg and 1821 ± 784 pg/mg in the Fn-n +handling and Fn-n + Mstim groups, respectively. 

The amount of NGF-protein in the intact LLS was 61 ± 22 pg/mg, with 51 ± 21 pg/mg and 33 ± 13 pg/mg total protein in 6 rats, each 2 months after Fn-n + handling, and Fn-n + Mstim, respectively.

### 7.4. Conclusions

Recovery of motor function after facial nerve injury and manual stimulation of the denervated and reinnervated muscles of the whisker pad is not associated with alterations in the expression of lesion-associated neurotrophic factors FGF2, IGF1 and NGF in the denervated muscles at an end-point analysis of 2 months after surgery. This finding does not exclude alterations in the trophic factors expression, which might have taken place in the musculature earlier after nerve injury.

## 8. Conclusions and Future Directions

The present review provides evidence that manual stimulation of denervated facial muscles can “override” the effects of the robust and consistent [89,136] but inappropriate [79] axonal regrowth in the target muscles, by reducing the degree of polyinnervation of reinnervated muscle endplates. This effect is apparently sufficient for restoration of motor function, and provides a platform for future behavioral, electro-physiological and morphological studies on motor recovery after facial nerve injury.

Unfortunately, the molecular basis of appropriate muscle target reinnervation and recovery of motor function remains enigmatic. One possible method of elucidating which mechanism(s) govern the regrowth of axons to their muscle targets with less polyinnervation of the motor endplates could be the performance of transcriptome analyses by means of mRNA sequencing in de- and re-innervated muscles in a time course manner. A recent study on the injured nerve has provided very interesting and promising results [152].

## Figures and Tables

**Figure 1 ijms-23-15101-f001:**
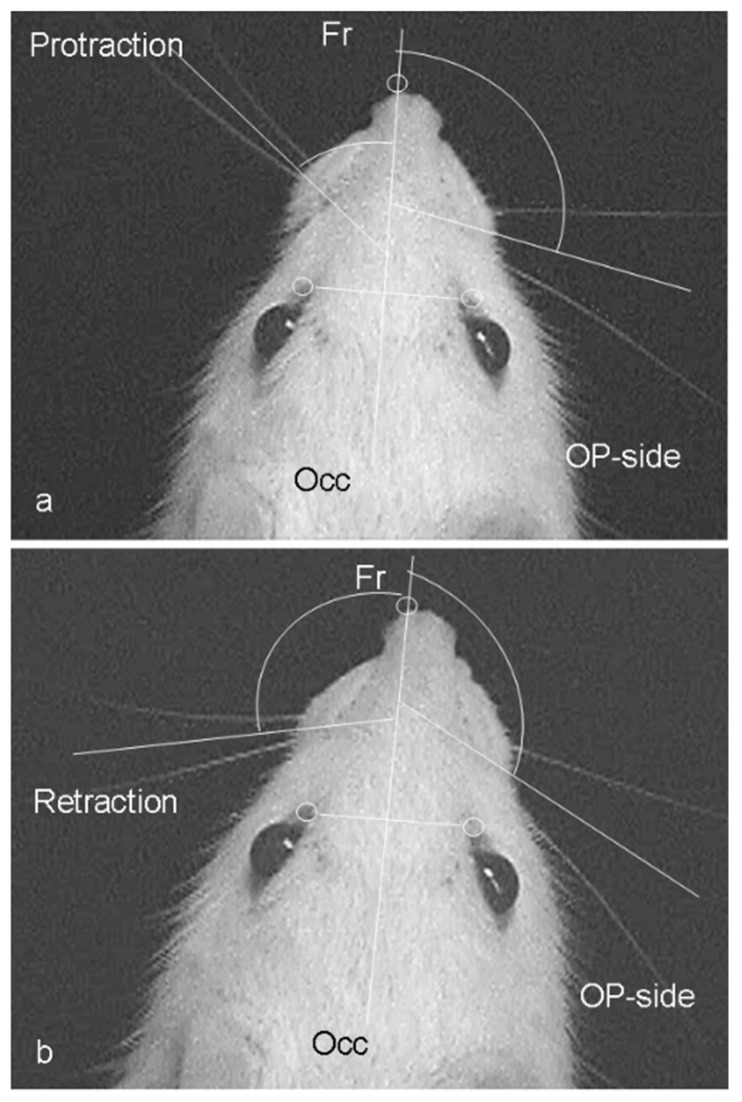
Video-based motion analysis of vibrissae motor performance on the left unoperated and right operated sides (OP-side) of the rat. Angles, angular velocity, and angular acceleration were measured during vibrissal protraction (**a**) and retraction (**b**). Note the significant change in angle from a sagittal line [Fr-Occ, which connects the frontally located tip of the nose (Fr) with the occipital bone (Occ)] during protraction and retraction on the intact side. The vibrissae on the operated side remain spastic (From Guntinas-Lichius et al., 2002) [59].

**Figure 2 ijms-23-15101-f002:**
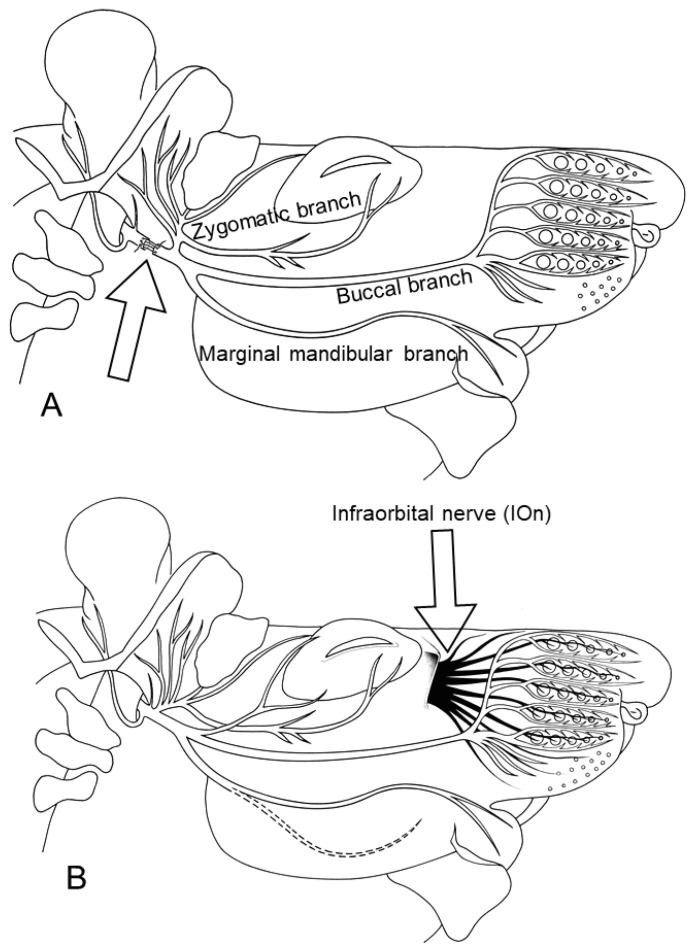
Schematic drawings illustrating the infratemporal portion of the facial nerve (Fn), and (**A**) the site of its transection and suture in one experimental rat group, as indicated by an arrow and (**B**) the close relationship between the peripheral fascicles of the Fn and those of the infraorbital nerve (IOn). The arrow shows the site of excision of the IOn in the experiments described in Section 4, demonstrating that afferent connections are important in improving functional recovery of whisking behavior and reducing polyinnervation of the motor endplates. The drawings are from Pavlov et al. (2008) [41].

**Figure 3 ijms-23-15101-f003:**
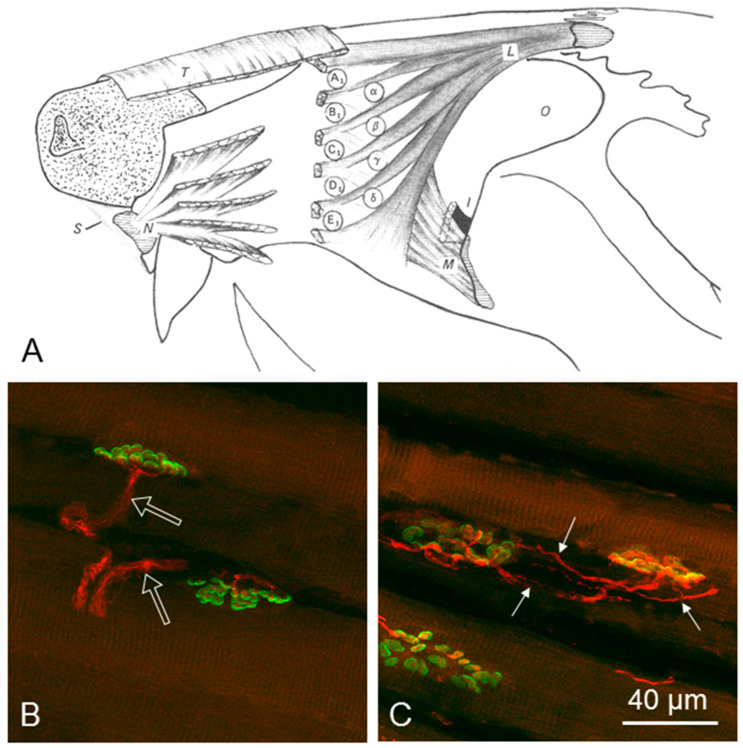
(**A**) Schematic drawing of the extrinsic vibrissae muscles according to Dörfl (1982) [74]: **α-δ**: the four caudal hair follicles, the muscles slings of which “straddle” the five vibrissae rows (**A_1_**–**E_1_**), **T**, transversus nasi muscle; **L**, levator labii superioris muscle; **N**, nasalis muscle; **M**, maxilolabialis muscle; **O**, orbit; **S**, septum intermusculare. Panels (**B**,**C**) show examples of a monoinnervated and a polyinnervated endplate, respectively, where superimposed stacks of confocal images of endplates in the levator labii superioris muscles of intact and surgically treated rats were visualized by staining the motor endplates with Alexa Fluor 488 α-bungarotoxin (green fluorescence), and immunostaining of the intramuscular axons for neuronal class III β-tubulin (Cy3 red fluorescence). Three axonal branches (arrows in (**C**)) reach the boundaries of the polyinnervated endplate delineated by the alpha-bungarotoxin staining. In contrast, the normally monoinnervated endplates in (**B**) are contacted by a single axon (empty arrows) with several preterminal rami, whilst in (**C**) the polyneuronally innervated endplate is contacted by three axonal branches. The drawing is from Sinis et al. (2009) [75].

**Figure 4 ijms-23-15101-f004:**
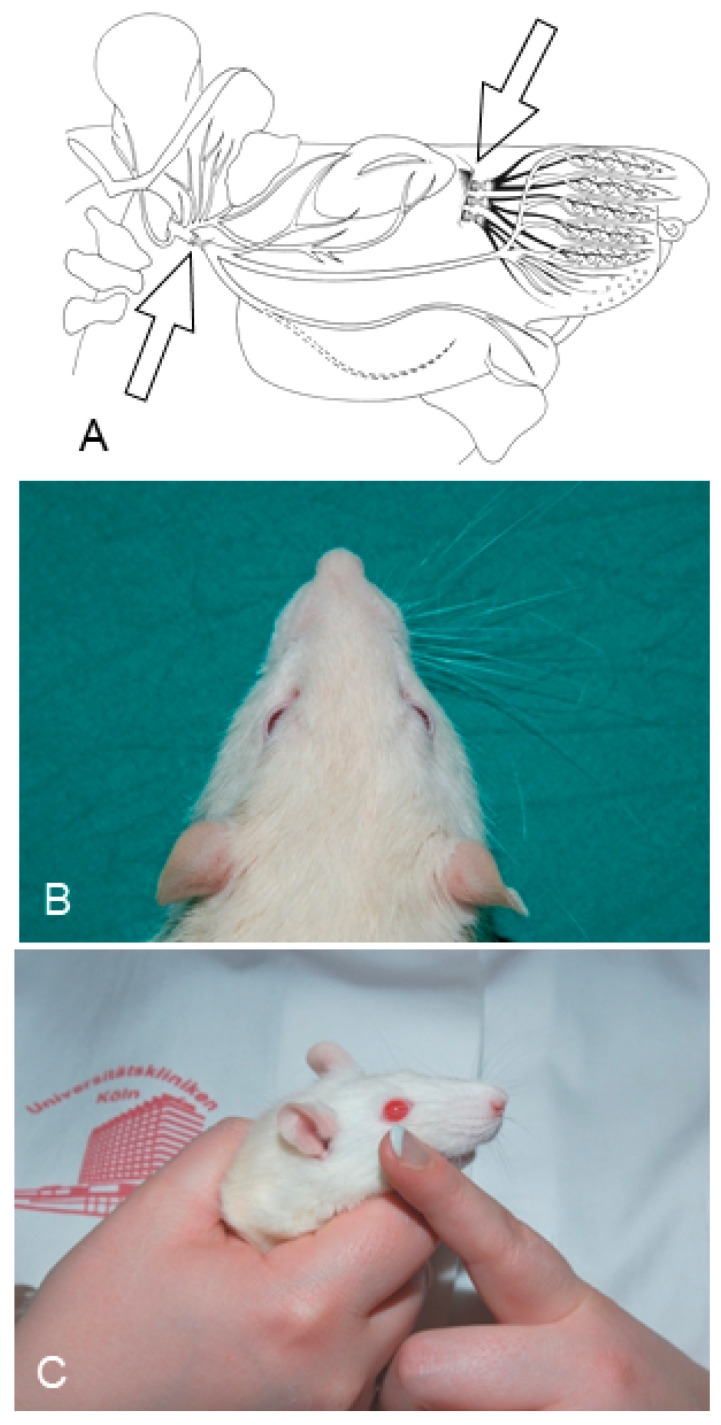
Schematic drawings illustrating transection and end-to-end suture of the infratemporal portion of the right motor facial nerve (Fn-n; (**A**), lower arrow) and the sensory infraorbital branch of the trigeminal nerve, IOn-n ((**A**), upper arrow), adopted from (Dörfl 1985; Semba and Egger 1986) [66,76]. (**B**,**C**) Images of postoperative treatments: (**B**) The vibrissae on the left side of the face were trimmed to maximize vibrissal use on the operated, right side; (**C**) Manual stimulation of the whisker pad skin and musculature on the operated, right, side. (**A**–**C**): From Bendella et al. (2011) [77].

**Figure 5 ijms-23-15101-f005:**
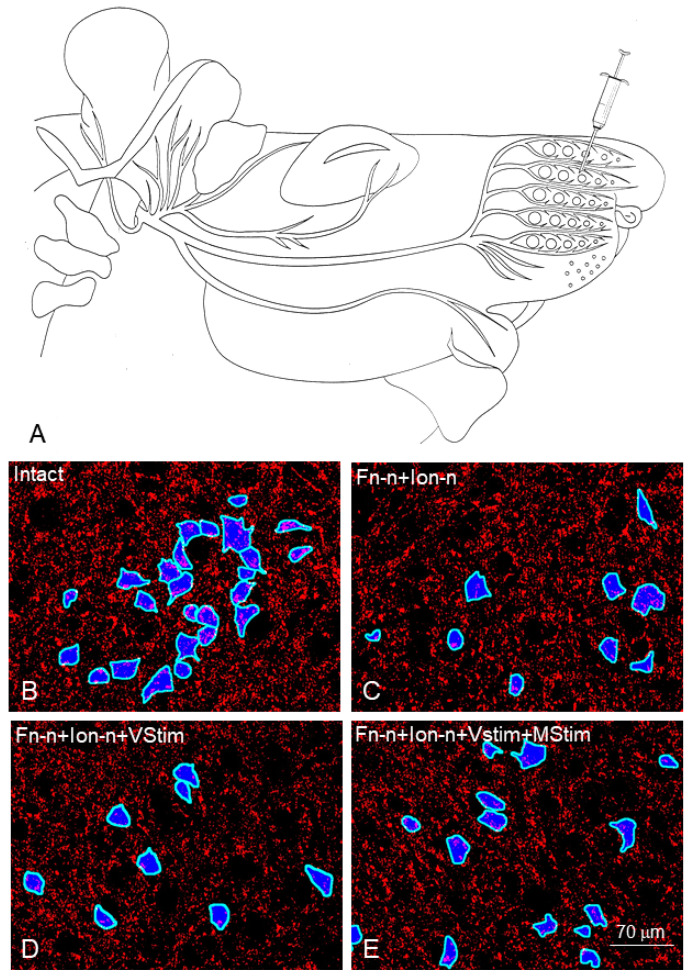
(**A**): Schematic drawing indicating the injection site of the retrograde tracer Fast Blue (FB; syringe) into the whisker pad. (**B**–**E**): Retrogradely labelled motoneuronal perikarya (blue) and synaptophysin-CY3 immunostaining of axosomatic nerve boutons in the intact facial nucleus (**B**), four months after Fn-n + Ion-n (**C**), after Fn-n + Ion-n + Vstim (**D**) and after Fn-n + Ion-n + Vstim + Mstim. FB-images were used to define “regions of interest” (RoIs) in each picture of the facial nucleus (ImageJ Software v1.38, NIH, Bethesda, MD, USA) through the following steps: (1) The dynamic range of the FB-images was maximized using gamma correction (γ = 0.2); (2) Images were sharpened by subtraction of a blurred copy (Gaussian blurring radius = 75 px); (3) Images were automatically thresholded using the Otsu algorithm to produce binary black and white images; (4) Motoneurons were included by selecting only FB-labelled areas with a value equal or greater than 500 µm^2^. (5) The resulting masks were used to measure the perimeter and area of the selected motoneurons. The perimeters were drawn and expanded in and out by 2 px, which generated RoIs from the closest perisomatic vicinity of the motoneurons with a width of 5 px (≈4 µm). All synaptophysin-positive profiles found within each of the predefined perisomatic RoIs of the thresholded images were counted, and the “numbers of perisomatic synapses per motoneuron” determined. From Bendella et al. (2011) [77].

**Figure 6 ijms-23-15101-f006:**
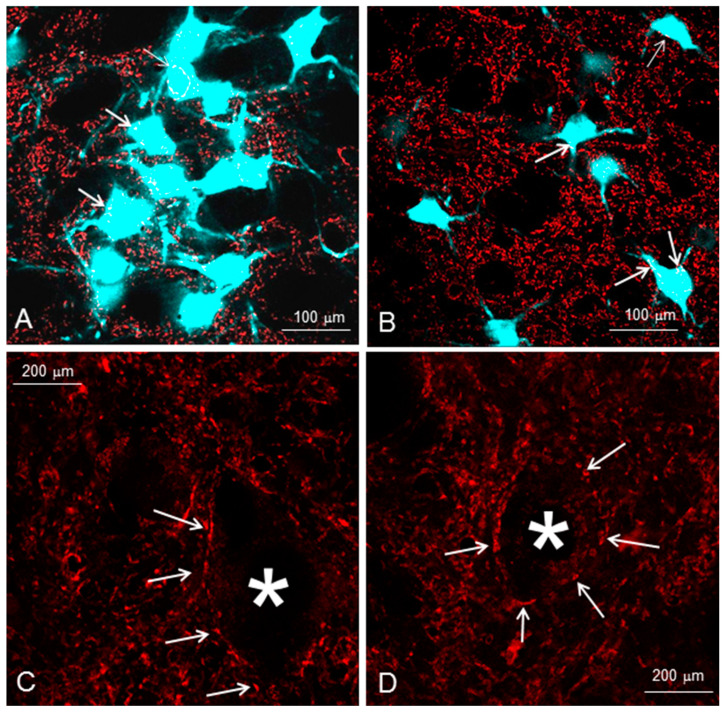
Analysis of perisomatic puncta. Low-power confocal images (1-µm-thick optical slices) show the appearance of synaptophysin-positive puncta (arrows) around intact (**A**) and axotomized (**B**) facial motoneurons, identified by the retrograde labeling with FB. High-power confocal images used for counting: (**C**) intact and (**D**) axotomized motoneuronal cell bodies (*), four months after regeneration following facial and infraorbital nerve transection and end-to-end suture. Immunostained sections through the facial nucleus were examined under a fluorescence microscope. Stacks of images of 1 µm thickness were obtained on a TCS SP5 confocal microscope (Leica) using a 40 × oil immersion objective and digital resolution of 1024 × 1024 pixels. Four adjacent stacks (frame size, 115 × 115 µm) were obtained consecutively in a rostrocaudal direction to sample more motoneurons. One image per cell at the level of the largest cell body cross-sectional area was used to count the number of perisomatic puncta. Motoneurons were easily identified by the retrograde labeling with fast blue (FB). Areas and perimeters were measured using the Image Tool 2.0 software program (University of Texas, San Antonio, TX, USA). Linear density was calculated as the number of perisomatic puncta per unit length. Between 105 and 120 cells were analyzed per group and parameter. From Bendella et al. (2011) [77].

**Figure 8 ijms-23-15101-f008:**
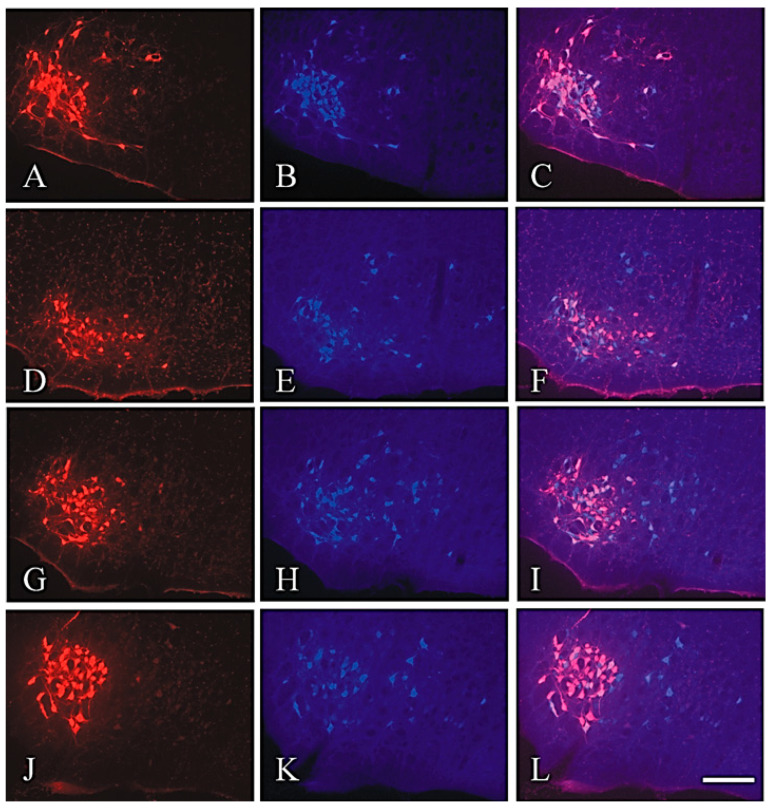
Rat brainstem 28 days after surgery on the buccal branch of the facial nerve (Bn). The lateral facial subnucleus, indicated by the pre-operative FG labelling, is in the left part of each picture. All photographs in the right column were produced by double exposure. (**A**–**C**) Intact facial nucleus with preserved myotopic organization of the motoneurons. Employing the selective filters, we depicted all pre-operatively FG-labelled (**A**) and all postoperatively FB-labelled (**B**) motoneurons. (**C**) In the intact facial nucleus, the proportion of double-labelled (FG + FB, pink to bright purple in colour) motoneurons is about 90%. (**D**–**F**) Lesioned facial nucleus 28 days after Bn-n. Whereas all preoperatively FG-labelled motoneurons are localized in the lateral facial subnucleus (**D**), those labelled postoperatively with FB are observed also in the intermediate facial subnucleus (**E**). Our quantitative estimates show that only about 27% of these FB-labelled motoneurons are double labelled (**F**) and belong to the original motoneuronal pool of the whisker pad. (**G**–**I**). Lesioned facial nucleus of a rat 28 days after Bn-n + IOn-ipsi-ex. All preoperatively FG-labelled motoneurons are in the lateral facial subnucleus (**G**). The postoperatively FB-labelled motoneurons are found in the lateral and intermediate facial subnuclei (**H**). The double-exposure picture (**I**) is similar to that in F, showing that about 32% of the FB-labelled cells were also FG labelled. (**J**–**L**). Lesioned facial nucleus of rat 28 days after Bn-n +IOn-contra-ex. All preoperatively FG-labelled motoneurons are in the lateral facial subnucleus (**J**) and some postoperatively FB-labelled cells are found in the intermediate facial subnucleus (**K**). Our counts show that after this type of combined surgery, the proportion of the double-labelled motoneurons (**L**) increased significantly to 41%. 50 µm thick vibratome sections; the scale bar indicates 100 μm. From Skouras et al. (2002) [91].

**Figure 9 ijms-23-15101-f009:**
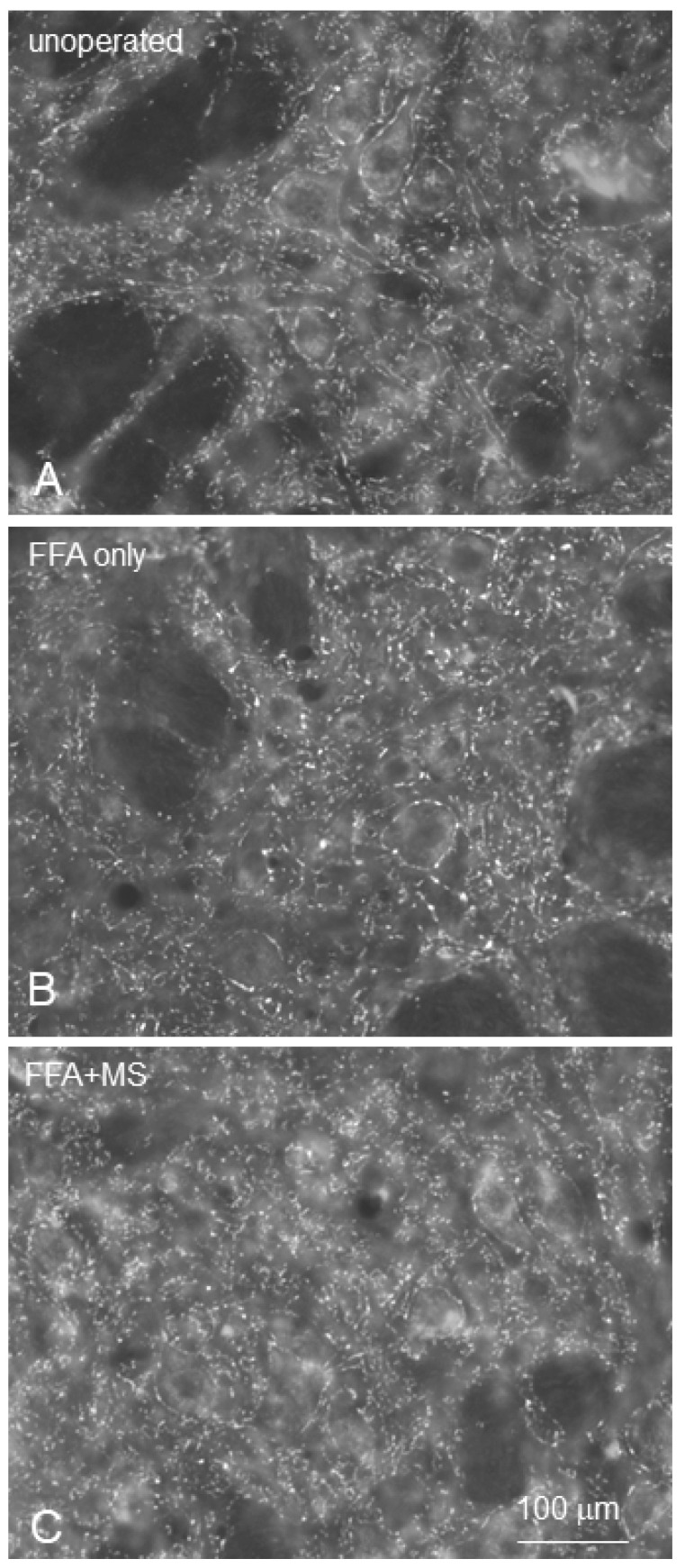
Immunostaining for synaptophysin in 30-μm thick vibratome sections from the facial nucleus in intact rats (**A**), in rats 2 months after Fn-n and handling (Fn-n + handling; (**B**)) and in rats that received Mstim of the vibrissal muscles after Fn-n (Fn-n + Mstim; (**C**)). Note the clearly discernible numerous puncta within the neuropil and around motoneuronal cell bodies in the facial nucleus representing synaptic terminals. From Pavlov et al. (2008) [41].

**Table 1 ijms-23-15101-t001:** Rat control and experimental groups, measurements (means + standard deviations) for experiments and measurements for behavior (whisking amplitude), muscle innervation including polyneuronal innervation, perisynaptic sensory terminal density on motoneurons and neurotrophic factor protein.

**Section 2** **(2 months)**	**Whisking** **Amplitude**	**NMJ** **Polyinnervation**		
**Rat groups**				
INTACT	62 ± 6°	0%		
Fn-n	19 ± 6°	53 ± 10%		
Fn-n + Mstim	51 ± 19°	22 ± 3%		
Fn-n + IOn-ipsi-ex	22 ± 3°	43 + 9%		
Fn-n + IOn-ipsi-ex + Mstim	14 ± 6°	51 ± 10%		
**Section 3** **(4 months)** **Rat Groups**	**Whisking** **Amplitude**	**NMJ** **Polyinnervation**	**Facial Motoneurons Perisomatic Terminal Fractional Area**	
INTACT	62 ± 6°	0%	17 ± 2%	
Fn-n + IOn-n	11 ± 4°	58 ± 8%	12 ± 1%	
Fn-n + IOn-n + Vstim	28 + 9°	40 ± 3%	13 ± 2%	
Fn-n + IOn-n + Mstim	30 ± 11°	40 ± 2%	13 ± 2%	
Fn-n + IOn-n + Vstim + Mstim	32 ± 10°	33 ± 10%	12 ± 2%	
**Section 4** **(4 weeks)**		**Buccal Branch Pathfinding**	**Buccal Branch Pathfinding**	**Buccal Branch Pathfinding**
**Rat groups**		**Superior-FG**	**Inferior-DiI**	**Sup + Inferior**
INTACT		91%	9%	0%
Bn-n		56%	21%	23%
Bn-n + ipsi IOn-ex		48%	39%	13%
Bn-n + contr IOn-ex		69%	23%	9%
**Section 5** **(4 weeks)** **Rat Groups**	**Accuracy of Whisker pad Reinnervation**			
INTACT	100%			
Bn-n	27%			
Bn-n + ipsi IOn-ex	32%			
Bn-n + contr IOn-ex	41%			
**Section 6** **(2 months)** **Rat Groups**	**Whisking Amplitude**	**LLS Polyinnervation of NMJ**	**Terminal Density (×10^6^) on Facial Motoneurons**	
INTACT	57 ± 13°	0%	34.3 ± 2.3	
Fn-n + Handling	19 + 6°	53 ± 10%	29.2 ± 1.8	
Fn-n + Mstim	51 ± 19°	22 ± 5%	33 ± 2.6	
**Section 7** **(2 months)** **Rat Groups**		**FGF2 Protein** **pg/mg**	**IGF1 Protein** **pg/mg**	**NGF Protein** **pg/mg**
INTACT		75 ± 16	1492 ± 87	61 ± 22
Fn-n + Handling		69 ± 24	2079 ± 300	51 ± 21
Fn-n + Mstim		56 ± 11	1821 ± 784	33 ± 13

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
