# Peer review of "Trigeminal Sensory Supply Is Essential for Motor Recovery after Facial Nerve Injury"

_ijms, 2022, doi:10.3390/ijms232315101_

Round 1

Reviewer 1 Report

The review addresses the clinically important problem of remaining functional impairment after facial nerve injury and repair. The core of the manuscript is the authors’ experimental studies on the rat facial nerve with some excursions into other, related fields of peripheral nerve regeneration. Although there are some information and discussion on general peripheral nerve regeneration issues, the material in the review is basically a systematic description of the authors’ own studies covering various manipulations to improve functional precision in whisker muscle reinnervation after facial nerve injury. The review appears as series of extended, and mostly quite detailed summaries, of these studies, terminating with the main conclusion on the critical role of sensory input to the injured facial motor nucleus.

Comments

1. The authors have presented their main findings, including the bottom line of this review, in a review paper 2019 (not cited). The current review appears mostly as a repeat of this review, although there are some additional background and discussion sections The pictures are mostly the same as in the previous review. The bulk of the current review is thus , and the text should have focused on novel aspects from their own studies (if there are any), and a more updated discussion on recent literature of peripheral nerve regeneration.

2. The authors include 146 citations, but a major part of the literature cited is typically from back in time. There is only one paper from 2021 (a clinical review), and none from 2020, despite the fact that the general problem discussed in the manuscript is still in a very active phase. The absence of newer literature can make the reader believe that nothing relevant has happened for several years. I am surprised that no recent clinical literature on Bell’s palsy is included.

3. A related question is whether selected citations actually support unreserved author statements. A few examples: P. 2, line 101; “2) trans-axonal exchange of abnormally intensive nerve impulses between axons from adjacent fascicles [9]”. This single reference from 1975 shows possible ephaptic transmission, but certainly no proof as the authors statement suggests. P. 2, line 103; “(3) alterations in synaptic input to facial motor neurons [10-12]” These are again old references (the most recent more than 25 years old) with suggestive data but no actual demonstration of the statement. Thus, the cited literature if far from convincing when it comes to these two explanations for the impairment after facial nerve injury repair. At the same time recent reviews in the field of facial nerve injury repair,  that might be valuable for the reader to informed about, are missing. E.g.,  Xie Y, Schneider KJ, Ali SA, Hogikyan ND, Feldman EL, Brenner MJ. Current landscape in motoneuron regeneration and reconstruction for motor cranial nerve injuries. Neural Regen Res. 2020 Sep;15(9):1639-1649. doi: 10.4103/1673-5374.276325; Yoo MC, Chon J, Jung J, Kim SS, Bae S, Kim SH, Yeo SG. Potential Therapeutic Strategies and Substances for Facial Nerve Regeneration Based on Preclinical Studies. Int J Mol Sci. 2021 May 6;22(9):4926. doi: 10.3390/ijms22094926. The latter is, by the way, published in the same journal as the submitted manuscript.

 4. It would be useful with a scheme of the network regulating facial motor neuron activity. The authors state (p. 3, line 124) that there is are ipsilateral monosynaptic connections on the motoneurons by the trigeminal afferent nerves [40–46]. However, this is most probably not correct (cf Deschênes M, Kurnikova A, Elbaz M, Kleinfeld D. Circuits in the Ventral Medulla That Phase-Lock Motoneurons for Coordinated Sniffing and Whisking. Neural Plast. 2016;2016:7493048. doi: 10.1155/2016/7493048). With the exception of the trigeminal motor nucleus, cranial nerve motor neurons are typically connected by second-order sensory neurons, mostly located in the reticular formation.

 5. The conclusion section should be “Conclusion and Future Directions”, and be boosted by at least some directions for future research, and not just iterate a plain conclusion from the results. Since the review could be a basis for further preclinical experiments, it would be helpful with suggestions on what molecular mechanisms might be relevant for a start.

Reviewer 2 Report

General comments 

The review by Rink-Notzon and colleagues is a good reflection of the Angelov group's (and others) experimental research on issues influencing functional recovery after facial nerve injury and will be of significant interest to both basic research scientists and clinicians. The review is well written (with a few exceptions that require clarification - see below) and well illustrated. As a general comment, the grouping together of much of the quantification for the various experiments into table 1 was frustrating, as this required going back and forward when reading the text. I suggest this information be split up and presented in its relevant sections. 

Specific comments 

Last sentence of Abstract (lines 27-29):  The authors state that "in all cases trigeminal nerve stimulation was beneficial ...". It would be helpful if the authors could clarify whether this means both vibrissal stimulation as well as manual stimulation of the whisker pad (since both would result in stimulation of the trigeminal nerve). 

Section 2 - Clinical importance (line 106-107): The authors state that axotomised facial motoneurons become hyperexcitable because of their "increased resting membrane potential and ... ". However, citation 13 states in the discussion that "In general agreement with Bradley et al. (1955), chromatolysed motoneurones have not significantly differed from normal motoneurones in respect of resting potential, spike potential and the after-potentials". The authors should state if this is a point of controversy.   

Line 139: "The research goals .... summarized in Table 1". The authors should add the research goals to each section of the table 1 as well as how many rats were used in each group of each experiment. Also, the temporal end points of data in Table 1 relating to whisker amplitude range from 2 months to 4 months (sections #3, #4 and #7). It might be more informative for the readers to see, in graphical form, the time course of recovery.    

Line 140: "We demonstrate that the electrical silence of axotomised facial motoneurons ..." should be modified because the authors do not demonstrate electrical silence of motoneurons.

Title of section 3 (lines 152-153): It would be more appropriate to wite that "Manual stimulation (Mstim) of the paralyzed AND DEAFFERENTED whisker pad worsens recovery of whisking". 

Figure 2: what is the significance of the branch of the marginal mandibular nerve that has been included in Fig2B but is not in Fig 2A? 

Lines 181-184: this sentence seems rather convoluted. The intended message would be clearer if delivered in a simpler grammatical form. Also, this sentence reports a worse functional outcome of manual stimulation of the whisker pad. It would be useful for the authors to include the p values where significant effects are claimed.

Section 3.2 would benefit from an explanation why the authors found it necessary to extrapolate morphological changes of the extrinsic vibrissal muscle as an indication of what may be taking place in the intrinsic vibrissal muscles.  This may be appreciated by the general reader because it has become widely appreciated that correlations of morphology and function are important for determining potential causative effects of certain experimental manipulations on outcome.

Title of section 3 (lines 152-153): It would be more appropriate to wite that "Manual stimulation (Mstim) of the paralyzed AND DEAFFERENTED whisker pad worsens recovery of whisking".  

Section 4 title (lines 218-219): For the general reader who may be unfamiliar with trigeminal nerve-facial nucleus circuitry, it may be useful to have an explanation how "removing the contralateral vibrissal hairs and massaging the ipsilateral whiskerpad" can be regarded as "Mild trigeminal indirect stimulation". 

Section 5 title (lines 339-340): Similar to the above, for the general reader who may be unfamiliar with trigeminal nerve-facial nucleus circuitry, it may be useful to have an explanation how "excision of the contralateral infraorbital nerve) after surgery on the buccal branch of the facial nerve" can be regarded as "Intensive trigeminal indirect stimulation". 

In the summary (lines 437-437): the authors state "excision of the contralateral but not the ipsilateral infraorbital nerve, reduced the collateral branching and/or increased elimination of the branches of regenerating nerve fibers after buccal nerve transection and surgical repair." It would be useful for the authors to state discuss what mechanism(s) might be causing this effect. 

Section 6 title (lines 440-441): Similar to the above title for section 5, please explain how "excision of the contralateral IOn after surgery on Bn"  can be regarded as "Intensive trigeminal indirect stimulation".

Discussion section (lines 534-535): Please re-write. The final term "at the earliest" is not good english and is confusing. 

Line 575: Please use a more appropriate term instead of "mighty" in  "Thus, it might be that the mighty ipsilateral afferent input...". 

Section 7 (Lines 627-629): please re-write this sentence. Also provide some explanation of how calculations of the total number of synaptophysin pixels per frame took into account large and randomly positioned areas of frames occupied by motoneuronal cell bodies of varying dimensions. Was there some form of compensation to ensure that frames from all groups didn't contain any bias? 

Section 7.3.1. Recovery of whisking (lines 637-641): It would be very informative to see the temporal pattern of recovery of whisker function in these groups in graphical form, rather than with single end-point data in a table.  Also, although the authors mention "speed of protraction" in the results - there is no data presented. 

Section 7.3.2. Loss of synapses in the lesioned facial nucleus: The authors document a small, but statistically significant reduction of synaptic density following Fn-n+handling. It would be informative to see this data more informative to the reader for this effect to be also be demonstrated as % of intact or control values.  Also, Table 7 refers to values for polyinnervation that are mentioned in the results section, but is referred to in the discussion (lines 667-668, also 697-698). Please ensure that all relevant topics are dealt with appropriately in each section of the review.

Line 679: "Intraorbital" should be infraorbital.

Section 8 Lines 717-720): Having cited their earlier work on growth factor mRNA and protein expression (ref 146),  they suggest looking at the "same neurotrophic factors" in the present review - but this is incorrect is incorrect since BDNF and IGF-2 were not included in the present review study. Also, the authors planned to correlate functional recovery of function with reduced polyneuronal innervation but show no data about this.  

Section 8.2 Methods: As mentioned earlier, animal numbers should be included in the methodology or results tables. The authors do mention "6 rats each" (line 741) but was this number part of a sub-group of animals used or was it the total number of animals used per group. The reader should not have to guess the authors' meaning.

Lines 735-736: The authors state "The transcription of the proteins FGF2, IGF1 and  NGF was determined for each sample in duplicate using sandwich ELISA-Kits". The authors surely mean translation.

Section 8.3 Conclusion (Lines 750-753): The authors state that "Recovery of motor function after peripheral nerve injury and manual stimulation of the denervated and reinnervated muscles is not associated with alterations in the expression of lesion-associated neurotrophic factors FGF2, IGF1 and NGF in the denervated muscles. Since the data presented is an end-point analysis at 8 weeks, it is impossible to make any statement about growth factor levels during the intervening weeks (or is this data also available but not demonstrated).  

Round 2

Reviewer 1 Report

The authors have provided acceptable explanations to my criticisms and revised the manuscript accordingly. 

Reviewer 2 Report

The authors are to be commended for going above (and beyond) what was required for clarification of a number of points.  

The inclusion of the flow charts has led to the unintended inclusion of additional boring mistakes which can be easily rectified:

Table 4 - day 75 column: (not 10 days later).

Table 5 - day 75 column: (also not 10 days later)

Table 7 - day 58-67 column. "The transcription of proteins ....." should be "the translation of proteins ...." 

This oversight was also repeated in the authors' response to my comment 22. In the methods section of part 8.2, the term "transcription" was used instead of translation. The authors' response to my comment was that this mistake was immediately corrected. This is not the case; the term "transcription" still remains uncorrected in the text.       
